# Hsp multichaperone complex buffers pathologically modified Tau

Antonia Moll[1,2], Lisa Marie Ramirez[1,2], Momchil Ninov [3,4], Juliane Schwarz[3,4], Henning Urlaub [3,4] &
Markus Zweckstetter [1,2✉]

Alzheimer's disease is a neurodegenerative disorder in which misfolding and aggregation of pathologically modified Tau is critical for neuronal dysfunction and degeneration. The two central chaperones Hsp70 and Hsp90 coordinate protein homeostasis, but the nature of the interaction of Tau with the Hsp70/Hsp90 machinery has remained enigmatic. Here we show that Tau is a high-affinity substrate of the human Hsp70/Hsp90 machinery. Complex formation involves extensive intermolecular contacts, blocks Tau aggregation and depends on Tau's aggregation-prone repeat region. The Hsp90 co-chaperone p23 directly binds Tau and stabilizes the multichaperone/substrate complex, whereas the E3 ubiquitin-protein ligase CHIP efficiently disassembles the machinery targeting Tau to proteasomal degradation. Because phosphorylated Tau binds the Hsp70/Hsp90 machinery but is not recognized by Hsp90 alone, the data establish the Hsp70/Hsp90 multichaperone complex as a critical regulator of Tau in neurodegenerative diseases.

[1] German Center for Neurodegenerative Diseases (DZNE), Von-Siebold-Str. 3a, 37075 Göttingen, Germany. [2] Department for NMR-based Structural Biology, Max Planck Institute for Multidisciplinary Sciences, Am Fassberg 11, 37077 Göttingen, Germany. [3] Max Planck Institute for Multidisciplinary Sciences, Bioanalytical Mass Spectrometry Group, Am Fassberg 11, 37077 Göttingen, Germany. [4] University Medical Center Goettingen, Institute of Clinical Chemistry, Bioanalytics, Robert-Koch-Strasse 40, 37075 Göttingen, Germany. ✉email: Markus.Zweckstetter@dzne.de

Tauopathies, among which Alzheimer's disease is the most prevalent form, are a class of devastating neurodegenerative disorders in which the microtubule-associated protein Tau misfolds and aggregates into insoluble deposits in the brain of patients[1,2]. Insoluble deposits of hyperphosphorylated Tau are closely associated with neurodegeneration and cognitive impairment[3]. Multiple lines of evidence link the pathogenic aggregation of Tau to changes in protein homeostasis and in particular to Hsp70 and Hsp90, two central members of the most abundant chaperone family of the heat shock proteins[4,5]. Little is known however about the orchestrated activities of molecular chaperones to counteract misfolding and aggregation of pathologically modified Tau.

Hsp70 and Hsp90 work together to promote the native state of globular clients beyond what can be achieved by the individual chaperones[6–8]. The joint action by the Hsp70/Hsp90 chaperone complex termed Hsp70/Hsp90 machinery recovers hormone-receptor interactions, regulates the activity of the tumor suppressor protein p53, enables a significant ATP binding blockage, assembles a functional RISC complex and enhances the kinetics of the hepatitis B virus reverse transcriptase[9–12]. In addition, the Hsp70/Hsp90 chaperone machinery plays a key role in cancer[13]: tumor Hsp90 is predominantly present in multichaperone complexes with Hsp90 displaying a specific high-affinity conformation for small molecules[12].

In vivo, Tau is post-translationally modified at multiple sites and with multiple modifications critically influencing Tau aggregation and Tau-induced neurodegeneration[2,14,15]. Ample evidence exists that hyperphosphorylated Tau accumulates during the development of Alzheimer's disease[1]. Tau phosphorylated at specific sites is actively investigated as biomarker for characterizing early phases of the disease[16,17]. In addition, Tau acetylation is linked to pathogenic aggregation and neurotoxicity[18], and the reduction of acetylated Tau is neuroprotective[19]. Post-translational modifications of Tau such as phosphorylation and acetylation might further play important roles in determining different Tau aggregate structures (strains) and thus underlie different tauopathies[20].

Here we show that the Hsp multichaperone complex formed by Hsp70, Hop, Hsp90, and p23 specifically regulates pathologically modified Tau and thus is a key target for fighting Tau misfolding and aggregation.

## Results

### Tau is a high affinity substrate of the Hsp70/Hsp90 machinery.
The core of the Hsp70/Hsp90 chaperone machinery comprises the complex of Hsp70 with Hop and Hsp90 (Fig. 1a). We reconstituted the Hsp70:Hop:Hsp90 complex from the purified individual chaperones in vitro and monitored complex formation by native page. Complex formation resulted in the attenuation of the bands of the free proteins and the appearance of a new band. Based on this analysis, the Hsp70:Hop:Hsp90 complex was formed when mixing the proteins in a molar ratio of 1:1:1 (Supplementary Fig. 1a). With increasing concentrations of the intrinsically disordered protein Tau, the intensity of the Hsp70:Hop:Hsp90 band decreased (Fig. 1b, c and Supplementary Fig. 1b, c). In parallel, an additional band appeared that contained the three chaperones and Tau (Fig. 1b, c and Supplementary Table 1). This is clear evidence that a so called 'client-loading complex', as previously described for globular substrates[21] exists for intrinsically disordered proteins.

The Hsp70:Hop:Hsp90:Tau complex was formed at different absolute protein concentrations (Supplementary Fig. 1c). Control experiments showed that neither the co-chaperone Hsp40, which is known to assist the Hsp70:substrate interaction[6], nor the addition of nucleotides were required to build up the Hsp70:Hop:Hsp90:Tau complex (Supplementary Fig. 1d) From the concentration-dependent intensity increase of the Hsp70:Hop:Hsp90:Tau complex band, we determined the apparent $K_D = 1.3 \pm 0.1$ µM (Fig. 1d). Given that native gel-based affinity determination is semi-quantitative, the $K_D$ values reported here should be regarded as apparent dissociation constants.

To define the domains of Tau that bind to the Hsp70/Hsp90 chaperone machinery, we used NMR spectroscopy. 2D $^{15}$N-$^1$H correlation spectra of Tau's backbone resonances were recorded in the absence and presence of the Hsp70:Hop:Hsp90 complex (Fig. 1e). Increasing concentrations of the Hsp70:Hop:sp90 complex induced changes in the Tau signals. Residues ranging from the proline-rich region P1/P2 to the flanking R' region were strongly broadened along with only few chemical shift perturbations (Fig. 1f and Supplementary Fig. 1e). Such changes are characteristic for residues interacting in the intermediate-to-slow exchange regime.

To investigate if Tau's repeat region alone is sufficient to evoke the binding to the Hsp70:Hop:Hsp90 complex, we measured the binding of the Tau construct K18, which only contains repeats R1-R4 (Supplementary Fig. 1f), to the Hsp70/Hsp90 multi-chaperone complex. With increasing concentrations of K18 only the level of unbound Hsp70 was reduced, whereas the amount of the Hsp70:Hop:Hsp90 complex remained the same (Supplementary Fig. 1g). This suggested that the isolated repeat region of Tau does not evoke an equally strong interaction with the Hsp70/Hsp90 chaperone machinery as full-length Tau.

We then made use of a second, longer Tau construct, termed K32, which comprises the repeats R1-R4 as well as the adjacent regions P2 and R' (Supplementary Fig. 1f). With increasing concentrations of K32, the formation of an Hsp70:Hop:Hsp90:K32 complex was observed (Supplementary Fig. 1g) along with the reduction of the unbound Hsp70:Hop:Hsp90 complex band, similar to the behavior of full-length Tau (Fig. 1c). Thus, the proline-rich region P2 and the pseudo-repeat R' contribute to a stable Hsp70/Hsp90 chaperone machinery:Tau interaction. In combination, the data show that the central part of Tau is the main interaction domain that associates with the Hsp70:Hop:Hsp90 complex. This interaction domain is predominantly positively charged and contains the hydrophobic repeats of Tau, which play a critical role in the misfolding and pathogenic aggregation of Tau[2].

### The co-chaperone p23 stabilizes the Hsp70/Hsp90 machinery:Tau interaction.
For full functionality of the Hsp70/Hsp90 chaperone machinery, the Hsp90 co-chaperone p23 is required[22]. To determine the importance of p23 for recognition of Tau by the human Hsp70:Hop:Hsp90 complex, we introduced p23 into the assembly reaction (Supplementary Fig. 2a–c). In the absence of Tau, the level of the Hsp70:Hop:Hsp90 complex remained unchanged with increasing amounts of p23 (Fig. 2b and Supplementary Fig. 2b). In contrast, when Tau was present, p23 associated with the Hsp70:Hop:Hsp90:Tau complex as evidenced by the appearance of a high molecular weight band located slightly above the band of the Hsp70:Hop:Hsp90:Tau complex (Fig. 2b). In parallel, the formation of the 5-component Hsp70:Hop:Hsp90:Tau:p23 complex decreased the amount of the ternary Hsp70:Hop:Hsp90 complex (Fig. 2b). The recruitment of p23 into the complex was further supported by the decrease of the amount of free p23 (Supplementary Fig. 2c, d). Similar to the Tau interaction with the Hsp70:Hop:Hsp90 complex (Supplementary Fig. 1d), the addition of different nucleotides had no effect on the binding behavior of p23 (Supplementary Fig. 2e).

To gain insight into the contribution of p23 to the recognition of Tau by the Hsp70/Hsp90 multichaperone complex, we recorded 2D $^{15}$N-$^1$H NMR correlation spectra of isotopically labeled Tau in

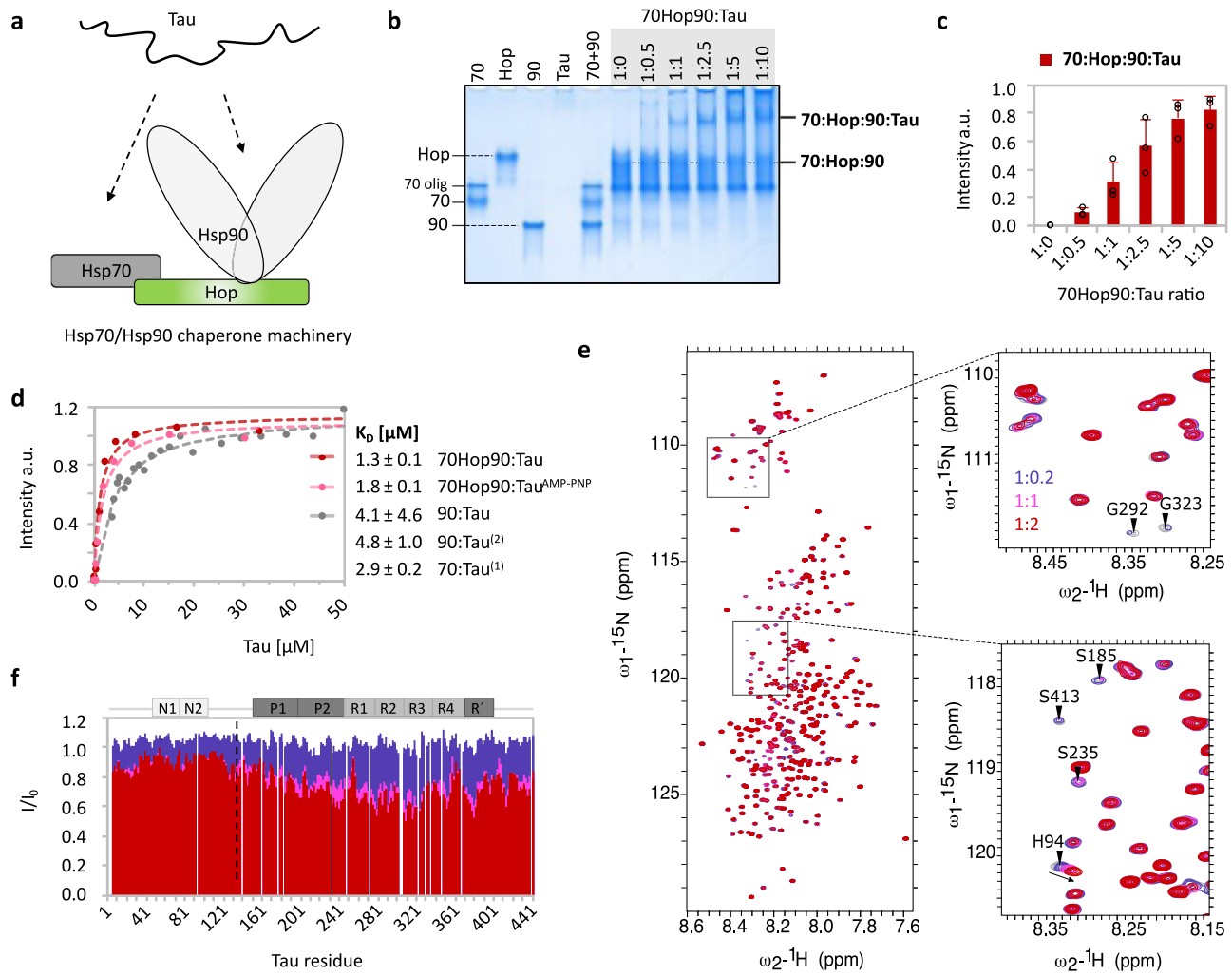

**Fig. 1 Tau is a high-affinity substrate of the Hsp70/Hsp90 chaperone machinery. a** Cartoon representation of Tau interacting with the Hsp70/Hsp90 chaperone machinery. **b** Native PAGE analysis of the Hsp70:Hop:Hsp90 complex (1:1:1 molar ratio) with increasing concentrations of Tau. Hsp70 and Hsp90 are abbreviated as "70" and "90", respectively. For Hsp70, a fraction of the protein was oligomeric (labeled as "70 olig"). **c** Quantitative analysis of the band intensities from (**b**). Data are presented as mean ± the standard deviation (SD) from three independent experiments. **d** Tau affinities for the Hsp70/Hsp90 chaperone machinery in the absence (red) and presence of AMP-PNP (rose), as well as for the individual chaperones Hsp70 and Hsp90 reported previously (2),(1)[40,64] and confirmed for the Hsp90:Tau interaction (gray). Errors represent SD from three independent experiments (70Hop90:Tau) or standard error of the nonlinear fit (70Hop90:Tau$^{AMP-PNP}$ and 90:Tau). **e** 2D $^{15}$N-$^{1}$H HSQC spectra of Tau alone (gray) and with increasing concentrations of the Hsp70:Hop:Hsp90 complex (1:1:1 molar ratio) using Tau:machinery molar ratios of 1:0.2 (purple), 1:1 (pink) and 1:2 (red). **f** NMR interaction profile observed in (**e**) (same color code). $I_0$ are the peak intensities of unbound Tau (gray spectrum in (**e**)). The black dotted line marks the distinction between fast (to the left) and intermediate-to-slow exchange (to the right) referring to low and high affinity, respectively. Tau domains are indicated on top. Source data are provided as a source data file.

the presence of the Hsp70:Hop:Hsp90 complex together with p23 (Supplementary Fig. 2f). Sequence-specific analysis showed that the peak broadening profiles of Tau bound to the Hsp70/Hsp90 chaperone machinery were highly similar in the absence and presence of p23 (Fig. 2c, top panel). Tau's central region remained the main interaction site. In addition, the N-terminal projection domain of Tau was largely unperturbed, i.e. it is highly flexible and freely accessible within both the 4-component Hsp70:Hop:Hsp90:-Tau and the 5-component Hsp70:Hop:Hsp90:Tau:p23 complex.

When compared to Tau's interaction profile with Hsp70 and Hsp90 alone (Fig. 2c, lower panel), the same residues of Tau were involved in either interaction. Hsp70 and Hsp90 binding involved four and five major dips/hot spots in the peak intensity profile of Tau, respectively. All five hot spots were also observed in the Hsp70:Hop:Hsp90:Tau and the Hsp70:Hop:Hsp90:Tau:p23 complex. The locally smaller intensity ratios in the presence of p23

(blue bars vs red line in Fig. 2c) suggest that the addition of p23 strengthens the interaction of Tau with the Hsp70/Hsp90 chaperone machinery. NMR experiments of Tau with p23 alone proved that p23 can directly bind to Tau, with most perturbations located in Tau's P2 region (Fig. 2d, lower panel). Similar chemical shift perturbations were observed for the Hsp70:Hop:Hsp90:-Tau:p23 interaction (Fig. 2d, top panel), in agreement with a direct interaction between Tau and p23 in the complex. The combined data reveal a high-molecular weight Tau:Hsp70/Hsp90 multichaperone complex.

**The Hsp70/Hsp90 multichaperone complex blocks Tau aggregation.** To investigate if the Hsp70/Hsp90 multichaperone complex regulates the pathogenic aggregation of Tau, we monitored Tau amyloid fibrillization kinetics in the absence and

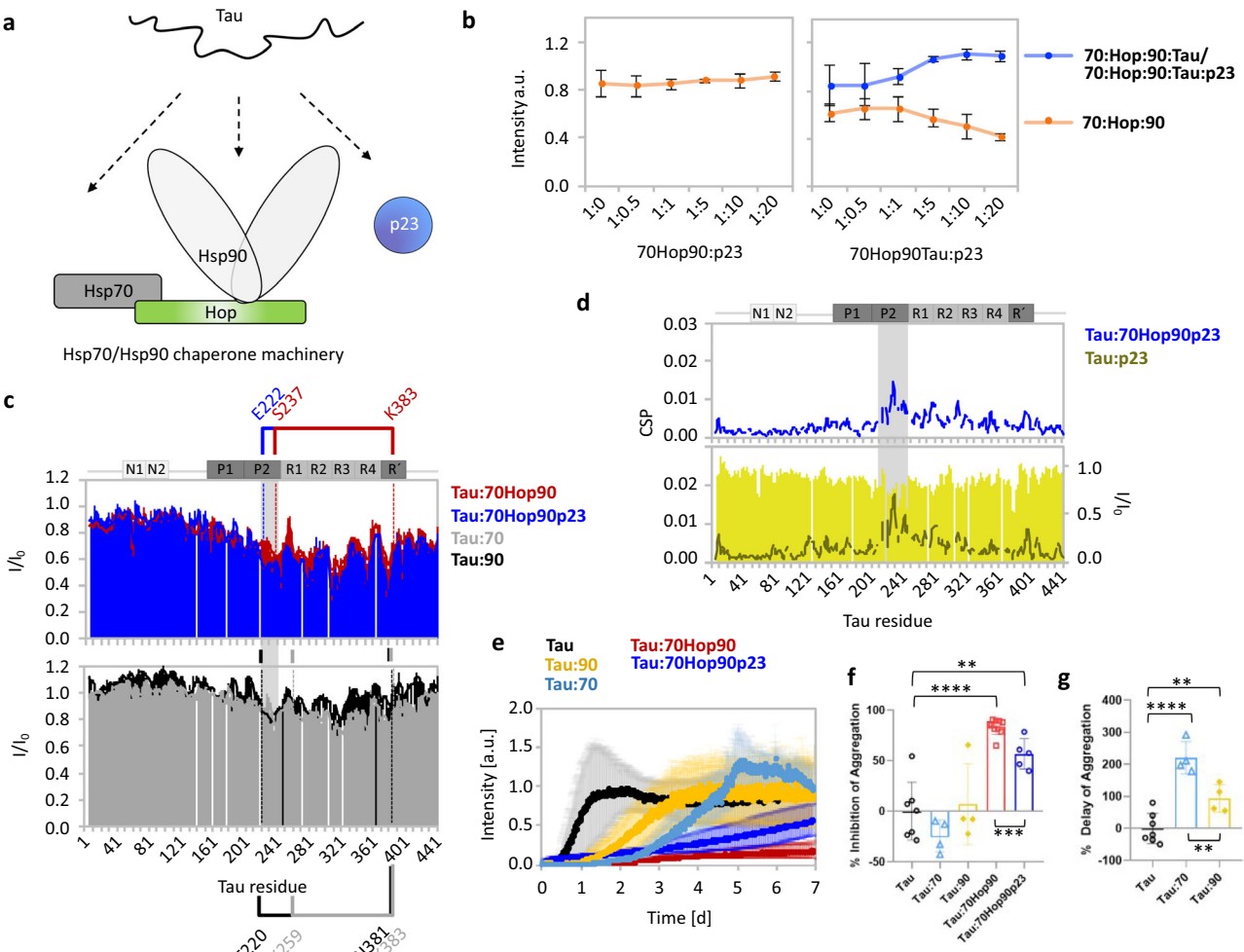

**Fig. 2 p23 stabilizes the Hsp70/Hsp90 multichaperone:Tau interaction. a** Cartoon representation of Tau interacting with the Hsp70/Hsp90 chaperone machinery and p23. **b** Quantitative analysis of the band intensities from Supplementary Fig. 2a, b demonstrating the formation of the 5-component Hsp70:Hop:Hsp90:Tau:p23 complex. The 4-component Hsp70:Hop:Hsp90:Tau and 5-component Hsp70:Hop:Hsp90:Tau:p23 complex were analyzed together due to the closely positioned bands. Data are shown as mean ± SD from three independent experiments. **c** NMR interaction profiles of Tau from Supplementary Fig. 2f in the presence of the Hsp70/Hsp90 chaperone machinery with (blue; top) and without p23 (red; top), and upon addition of Hsp70 (gray; bottom) and Hsp90 (black; bottom) only. Results using Tau:chaperone molar ratios of 1:2 are shown for all experiments. The main interaction sites are emphasized in brackets. **d** Top: chemical shift perturbations of Tau in presence of Hsp70, Hop, Hsp90 and p23. Bottom: NMR interaction profile of Tau in the presence of p23 (tenfold molar excess). **e** Aggregation assay of Tau alone and in the presence of individual or combined components of the Hsp70/Hsp90 chaperone machinery using a mole ratio of 1:0.2 (Tau:machinery or Tau:chaperone). For panels **e**–**g**, data are presented as mean values ± SD from n independent experiments: $n = 7$ for Tau, $n = 4$ for Tau:Hsp70, $n = 4$ for Tau:Hsp90, $n=10$ for Tau:Hsp70:Hop:Hsp90, $n = 5$ for Tau:Hsp70:Hop:Hsp90:p23. For panels **f**, **g**, null hypothesis testing was performed using an unpaired, two-tailed t-test. **f** Inhibition of Tau aggregation by individual or combined components of the Hsp70/Hsp90 chaperone machinery (*$p \leq 0.033$, **$p \leq 0.0021$, ***$p \leq 0.0002$, ****$p \leq 0.0001$) from panel e (also related to Supplementary Fig. 2g). **g** Delay of Tau aggregation by Hsp70 or Hsp90 (*$p \leq 0.033$, **$p \leq 0.0021$, ***$p \leq 0.0002$, ****$p \leq 0.0001$) determined from the ThT curves in panel e. The % delay cannot be accurately calculated for Tau:70Hop90/ Tau:70Hop90p23 because the ThT curves for these systems are almost flat, i.e. not sigmoidal. A lower limit for the % delay in these systems is ~200%, which is the % delay of Tau aggregation in the presence of Hsp70. Source data are provided as a source data file.

presence of the Hsp70:Hop:Hsp90 complex (Fig. 2e–g). So far, studies investigating the impact of molecular chaperones on Tau aggregation were complicated by the need of highly negatively charged co-factors to convert Tau into amyloid fibrils. We recently overcame this bottleneck and established a co-factor-free assay in which full-length Tau efficiently aggregates into amyloid fibrils[23].

We performed the co-factor-free aggregation assay using Tau with individual or combined components of the Hsp70/Hsp90 multichaperone complex. We quantified the % delay of Tau aggregation from the midpoint of the growth phase of the Tau fibrillization curves and determined % inhibition of Tau

aggregation from the maximal fluorescence of Thioflavin-T (ThT). In the conditions of the co-factor-free aggregation assay, only Tau but none of the chaperones or co-chaperones formed ThT-positive amyloid fibrils (Supplementary Fig. 2g). Addition of substoichiometric concentrations of the Hsp70 or Hsp90 delayed Tau aggregation but did not decrease the magnitude of the ThT-signal at the end of the aggregation assay (Fig. 2e–g). In contrast, addition of the Hsp70:Hop:Hsp90 complex resulted in an almost flat ThT profile, indicating that the machinery both inhibited and delayed Tau fibrillization (Fig. 2e, f and Supplementary Fig. 2g). The inhibitory effect was concentration dependent with up to ~98% inhibition at substoichiometric amounts of the machinery

(Supplementary Fig. 2g, h). Strong inhibition was likewise observed when p23 was present (Supplementary Fig. 2g, h).

**Molecular interactions in the human Tau:Hsp70/Hsp90 chaperone complex**. To gain insight into the molecular interactions that stabilize the Hsp70/Hsp90 chaperone machinery:Tau complex, we recorded methyl-transverse relaxation optimized spectroscopy (TROSY) NMR spectra[24] of Hsp90 when integrated into the Hsp70/Hsp90 machinery, the 4-component Hsp70:Hop:Hsp90:Tau as well as the 5-component Hsp70:Hop:Hsp90:Tau:p23 complex (Figs. 3, 4a, b). In presence of Hop and Hsp70 the signals of Hsp90 were broadened with additional chemical shift perturbations induced in Hsp90's NTD (I37, I75), MD (I334, I352) and CTD (I579, I627, I683) when compared to the addition of Hop only (Fig. 3a, b). By highlighting the perturbed residues in the previously demonstrated $Hsp90_2:Hop_1$ model[25] we could expose two Hsp70 binding pockets on Hsp90 as part of the Hsp70:Hop:Hsp90 complex (Fig. 3c). Our observation of signal broadening (far below 50%) for methyl protons along the length of Hsp90 (Fig. 3b) upon titration with Hop and Hsp70 indicate that the Hsp90 domains are potential binding sites of Hsp70 while Hsp90 is associated with Hop. This may be in agreement with an assembly in which a single Hsp70 molecule binds Hsp90:Hop via Hop[26], or one in which a single Hsp70 molecule is bound to each of the two Hsp90 molecules in the dimer as observed in a recent cryo-EM structure of a client-loading complex containing a $Hsp70_2:Hop_1:Hsp90_2$ machinery[21]. Notably, in the same cryo-EM study, a stoichiometry of $Hsp70_1:Hop_1:Hsp90_2$ was also observed[21].

In agreement with the high molecular weight of the multichaperone complexes, most Hsp90 signals were broadened beyond detection upon the addition of Tau and p23 (Fig. 4a). Only the signals of Hsp90's charged linker persisted indicating that this region remains highly mobile within the Hsp70/Hsp90 chaperone machinery (Fig. 4b). For residue I224 located in the charged linker, we observed peak splitting (Fig. 4a) suggesting that the charged linker of Hsp90 transiently interacts with the rigid core of Hsp70/Hsp90 multichaperone complex.

To further characterize the molecular interactions in the Hsp70:Hop:Hsp90:Tau:p23 complex, we performed chemical crosslinking with two different crosslinkers followed by liquid chromatography (LC)-mass spectrometry (MS)/MS (Supplementary Fig. 3a). We selected the 5-component complex, because of the additional stabilization provided by p23 (Fig. 2b, Supplementary Fig. 3b). The analysis identified an extensive network of intra- and intermolecular crosslinks (Fig. 4c, Supplementary Fig. 3c–e and Supplementary Data 1, Supplementary Data 2). The intermolecular crosslinks showed that Hsp70's SBD contacts the TPR1 domain of Hop, and Hop's TPR2B domain is in close proximity to the middle and C-terminal domain of Hsp90 (Fig. 4c, d and Supplementary Fig. 3e), in agreement with previous studies[27]. Zooming into Tau we find that its repeat region is the main interaction domain for each component of the Hsp70/Hsp90 chaperone machinery. Most crosslinks with the repeat region of Tau were present for Hsp70's SBD, Hsp90's CTD, and the TPR1 as well as TPR2B/DP2 domains of Hop (Fig. 4c, d and Supplementary Fig. 3e). In agreement with a direct Tau:p23 interaction demonstrated by NMR (Fig. 2d), the majority of p23 crosslinks were found with the P2 region of Tau (Fig. 4c, d and Supplementary Fig. 3e). Notably, the second half of the P2 region (residues~220–240, highlighted in gray) was shown by NMR to be the additional binding site for Hsp90 that is not involved in Hsp70 binding (Fig. 2c). This is suggested by the peak broadening of these Tau residues upon titration with Hsp90 ($I/I_0 < 1.0$), which is largely absent in the titrations with Hsp70 (most $I/I_0$ values in this region are close to 1.0).

To estimate the stoichiometry of the complex of Tau with Hsp70, Hop, and Hsp90, we used DSS-crosslinking and density gradient centrifugation followed by size exclusion chromatography (Fig. 4e, f). For the Hsp70:Hop:Hsp90 complex in the absence of Tau, we obtained an elution volume corresponding to a molecular mass of 411 kDa (Fig. 4e, f). In the presence of Tau, however, the estimated mass increased to 739 kDa (Fig. 4e, f), i.e. much larger than the molecular weights expected for a $Hsp70_2:Hop_1:Hsp90_2:Tau_1$ complex (425 kDa) and a $Hsp70_1:Hop_1:Hsp90_2:Tau_1$ complex (354 kDa). However, given the broad nature of the complex peak in SEC it cannot be excluded that multiple stoichiometries ranging from ~400 kDa to ~700 kDa may be formed (corresponding to elution volumes ~11 mL to ~9 mL). This is consistent with SDS-PAGE analysis of the cross-linked Hsp70:Hop:Hsp90:Tau complexes that show band smearing corresponding to molecular weights above 200 kDa (Supplementary Fig. 3b). The wide range of stoichiometries may correspond to 1:1 to 2:2 complexes of Tau:machinery, with possibly one or two Hsp70 units within the Hsp70:Hop:Hsp90 machinery. Notably, in the presence of p23 the main cross-linked complex shows a quite defined band in SDS-PAGE (Supplementary Fig. 3b) as well as a narrower peak in the SEC chromatogram (~700 kDa molecular weight corresponds to the elution volume at the peak maximum, Fig. 4e). This suggests that the addition of p23, which directly binds to Tau (Fig. 2d), further stabilized the high-molecular weight complex. The combined data point to the formation of high molecular complexes of Hsp70:Hop:Hsp90:Tau and Hsp70:Hop:Hsp90:Tau:p23 with variable stoichiometry.

**CHIP disassembles the Hsp70:Hop:Hsp90:Tau:p23 complex**. Proteins can be guided towards proteasomal degradation via a direct interaction of Hsp70 or Hsp90 with the E3-ubiquitin ligase CHIP (carboxyl terminus of Hsc70-interacting protein)[28,29]. CHIP is the functional counterpart of Hop competing for the binding to Hsp70 and Hsp90 via the TPR domain, and by that ensuring a continuous cycle of machinery buildup and breakdown (Fig. 5a). Consistent with the competition of CHIP and Hop for binding to Hsp70 and Hsp90, we observed the disintegration of the Hsp70:Hop:Hsp90:Tau complex when adding an equimolar concentration of CHIP (Fig. 5b, c left panel). Comparison with the native page behavior of individual Hsp70:CHIP and Hsp90:CHIP complexes in either the presence or absence of Tau further suggested that CHIP-mediated disintegration of the Hsp70:Hop:Hsp90:Tau complex leads to Hsp70:CHIP, Hsp70:CHIP:Tau, Hsp90:CHIP and Hsp90:CHIP:Tau subcomplexes (Fig. 5b). Notably, a smaller amount of Hop was released by CHIP from the 5-component Hsp70:Hop:Hsp90:Tau:p23 complex (Fig. 5d) when compared to the 4-component Hsp70:Hop:Hsp90:Tau complex, further supporting the stabilization of the Hsp70/Hsp90 machinery:Tau interaction by p23 (Fig. 4e, Supplementary Fig. 3b). The data demonstrate that by interacting with either Hop or CHIP, Hsp70 and Hsp90 can maintain a dynamic equilibrium between Tau retention via the Hsp70/Hsp90 chaperone machinery and CHIP-mediated Tau degradation (Fig. 5a). However, despite the roughly comparable affinities of Hop and CHIP for Hsp70 and Hsp90[30], with regard to the rather low intracellular concentrations of CHIP (0.0094 μM) compared to Hop (1.2 μM)[31], it remains to be elucidated to which extent a dynamic interplay between Tau retention via Hop and Tau degradation via CHIP is maintained by Hsp70 and Hsp90 in vivo.

**The Hsp70/Hsp90 multichaperone complex binds pathologically modified Tau**. To gain insight into the molecular underpinnings of the interaction of the Hsp70/Hsp90 multichaperone

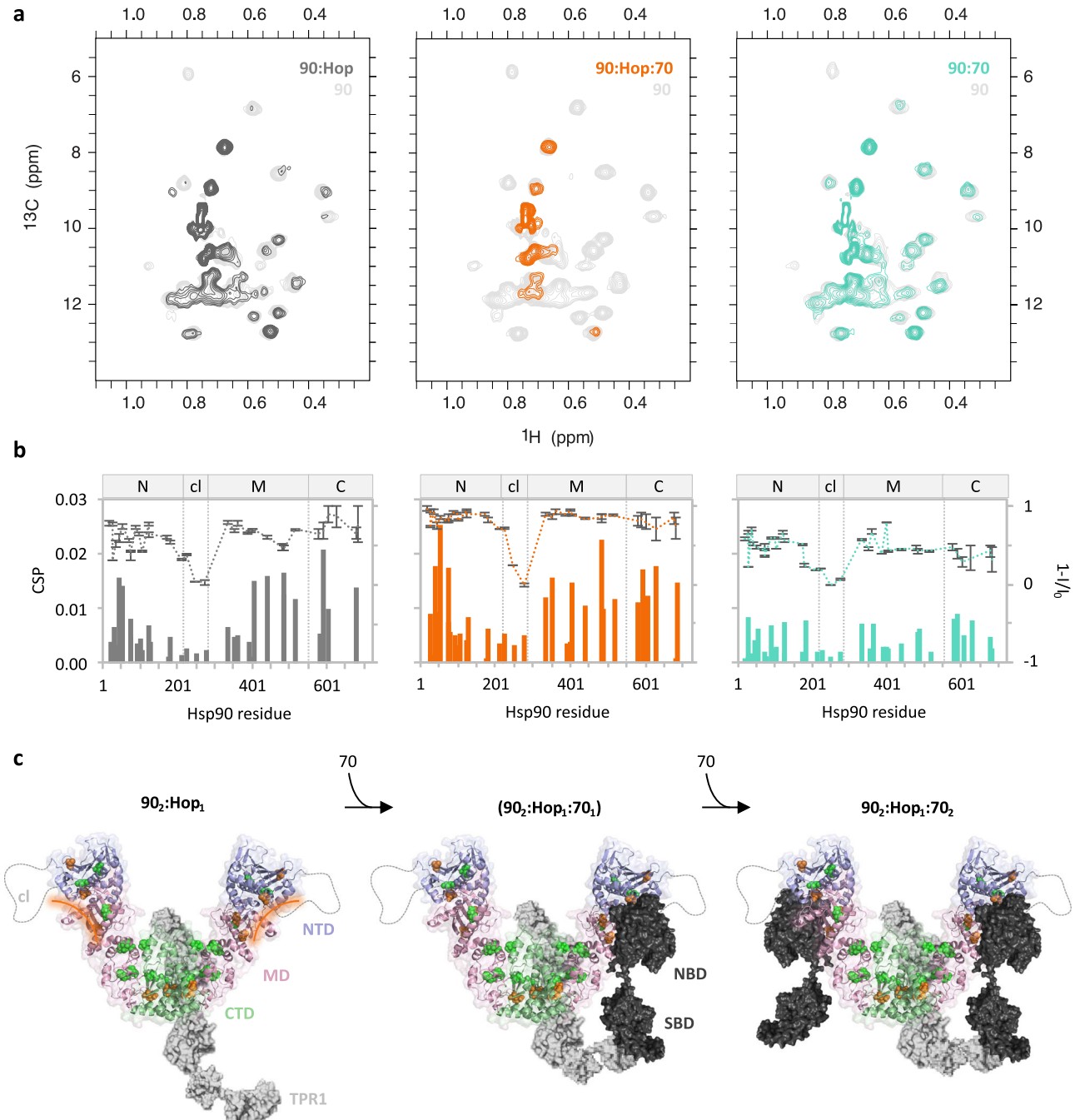

**Fig. 3 Formation of the Hsp70/Hsp90 multichaperone complex followed by NMR. a** 2D $^{13}$C-$^1$H methyl-TROSY spectra of Hsp90 alone (light gray) and in the presence of Hop and Hsp70 (orange). Spectra of Hsp90 with Hop only (dark gray)[25] and Hsp70 only (turquoise) are shown for comparison. **b** Chemical shift perturbation (bars) and peak intensity changes (dotted lines) of Hsp90's isoleucine δ-methyl groups observed in (**a** same color code). $I_0$ is the Hsp90 signal intensity in the unbound state (light gray spectra in **a**). Error bars were calculated from the signal-to-noise ratio of each spectrum ($n = 1$ independent experiment). Hsp90 domains are depicted on top of each plot. **c** Cartoon representation illustrating the formation of the Hsp70/Hsp90 chaperone machinery. Front view of the V-shaped Hsp90:Hop complex is shown (Hsp90 domains: N-terminal domain NTD – purple, charged linker cl – gray, middle domain MD – pink, C-terminal domain CTD – green; Hop is depicted in gray). Affected isoleucines in Hsp90 are displayed with spheres and colored in green and orange for the Hsp90:Hop and Hsp90:Hop:Hsp70 interaction. The potential binding site of Hsp70 in Hsp90's NTD and MD is highlighted with an orange line. One Hsp70 molecule is bound via its SBD to Hop's TPR1 domain[27] – the second Hsp70 molecule may bind the Hop-induced V-shape conformation of Hsp90[25,65] independent of Hop. Hsp70 is represented in black. NBD and SBD are Hsp70's nucleotide- and substrate-binding domain, respectively. Source data are provided as a source data file.

complex with pathologically modified Tau, we phosphorylated recombinant Tau in vitro by two different kinases (Cdk2 and MARK2) and also prepared separate samples in which the protein was acetylated with either the acetyltransferases CBP or p300. The phosphorylated (PTau) and acetylated Tau (AcTau) proteins remained monomeric in solution as evidenced by hydrodynamic radii in the range from 6.2–7.0 nm (Supplementary Fig. 4a)[32]. Native PAGE analysis showed that the proteins

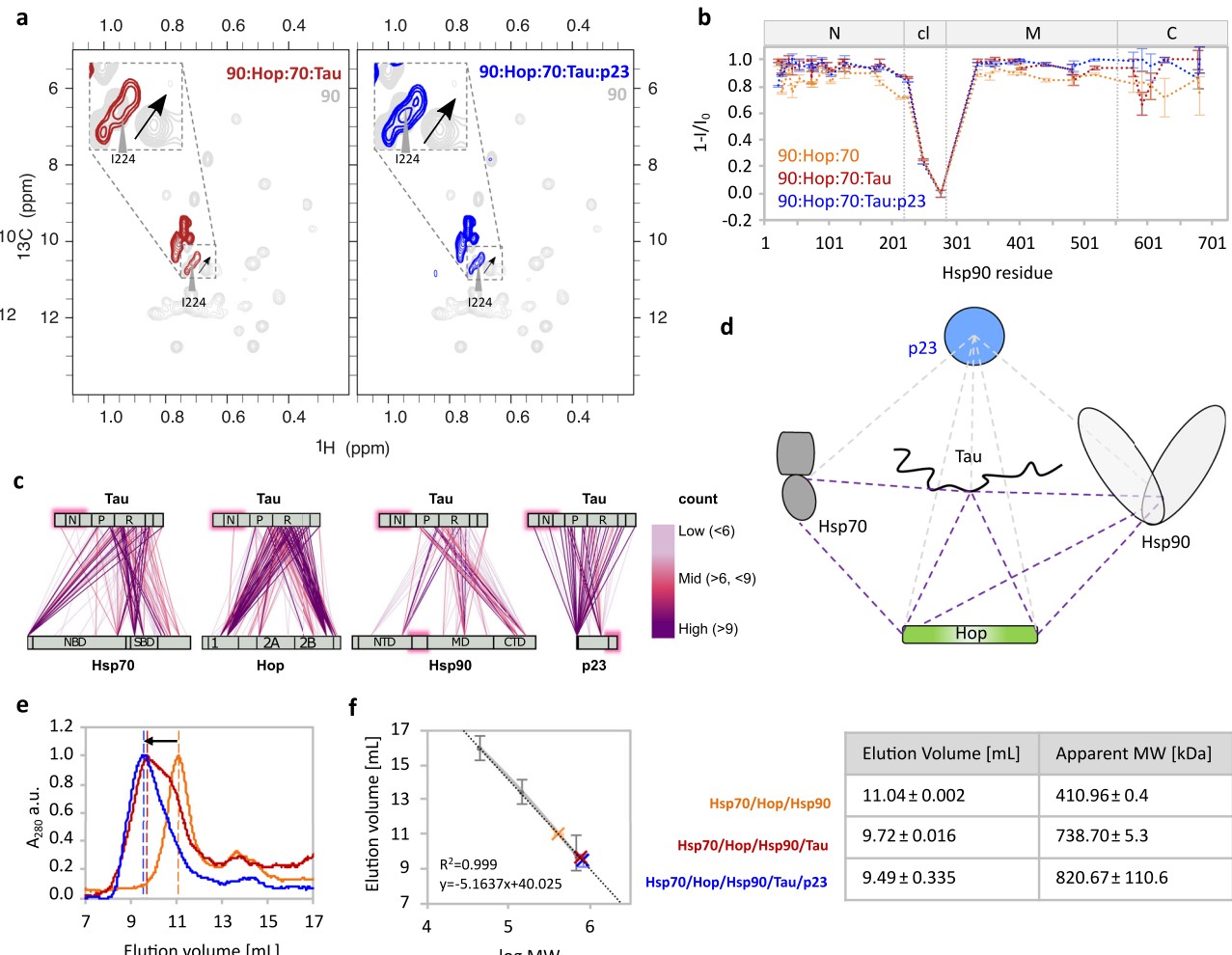

**Fig. 4 Molecular interactions within the Tau:Hsp70/Hsp90 multichaperone complex. a** 2D $^{13}C$–$^1H$ methyl-TROSY spectra of Hsp90 alone (light gray) and in the Hsp70:Hop:Hsp90:Tau (red) and the Hsp70:Hop:Hsp90:Tau:p23 complex (blue) with signal splitting of Ile224 (marked with a black arrow). **b** Peak intensity profile of Hsp90's isoleucine δ-methyl groups observed in **a** and (Fig. 3). Hsp90 domains are depicted on top. Error bars were calculated from the signal-to-noise ratio of each spectrum ($n = 1$ independent experiment). **c** DSS crosslink analysis of Tau within the Hsp70:Hop:Hsp90:Tau:p23 complex color coded based on the number of crosslinked peptide spectrum matches (CSMs). Count of CSM higher than 9 (purple), 6–9 (pink) and 3–6 (rose). Protein domains are indicated as gray bars; Hsp70: NBD-linker-SBD; Hop: TPR1-DP1-TPR2A-TPR2B-DP2; Hsp90: NTD-cl-MD-CTD; Tau: N1-N2-P1-P2-R1-R2-R3-R4-R'; p23: NTD-tail. The pink boxes mark regions likely to remain flexible. **d** Major intermolecular crosslinks from **c** connecting the Tau's repeat domain with Hsp70's SBD, Hsp90's CTD, Hop's TPR1 and TPR2B-DP2 and p23. **e** Size-exclusion chromatograms of the Hsp70:Hop:Hsp90 (orange), the Hsp70:Hop:Hsp90:Tau (red) and the Hsp70:Hop:Hsp90:Tau:p23 complex (blue). The black arrow marks the peak shift upon Tau binding. **f** Elution volumes from **e** plotted against the logarithmic molecular weight (log MW) based on a reference standard line (gray). Error bars for the standard curve represent the peak widths at half height. MW calculated from the elution volumes of two independent experiments are tabulated. Data for elution volumes and calculated MW of the tau/chaperone assemblies are reported as mean ± SD. Source data are provided as a source data file.

were heterogeneously modified (Supplementary Fig. 4b), complicating the analysis of complex formation by native PAGE due to band smearing. However, the amount of the ternary Hsp70:Hop:Hsp90 complex decreased with increasing concentrations of post-translationally modified Tau in all cases (Supplementary Fig. 4b), consistent with a loss of this complex due to the formation of Hsp70:Hop:Hsp90:PTau/AcTau complexes. For further analysis, we took advantage of the observation that the Hsp70:Hop:Hsp90:PTau$^{Cdk2}$ interaction resulted in a high molecular weight band that was distinct from the unbound PTau$^{Cdk2}$ and thus could be used for quantitative analysis.

Cdk2 generates a phosphorylation pattern in Tau that is overlapping with that observed in Alzheimer's disease (Supplementary Table 2, Fig. 6a)[33]. In particular, LC-MS/MS confirmed the phosphorylation of epitopes, which are recognized by the monoclonal antibodies AT8, AT180, and PHF1 detecting

phosphorylated Tau species in Alzheimer's disease[34]. When we titrated the Hsp70/Hsp90 multichaperone complex with increasing concentrations of PTau$^{Cdk2}$, we observed the formation of Hsp70:Hop:Hsp90:PTau$^{Cdk2}$ and Hsp70:Hop:Hsp90:PTau$^{Cdk2}$:p23 complexes comparable to those formed by unmodified Tau (Fig. 6b, and Supplementary Fig. 4b, c). In addition, we found that both PTau$^{Cdk2}$ and unmodified Tau interact with unbound Hsp70 (Supplementary Fig. 4d). In contrast, PTau$^{Cdk2}$ displayed a decreased affinity for Hsp90 when compared to unmodified Tau (Fig. 6b, c and Supplementary Fig. 4d). The decreased affinity of PTau$^{Cdk2}$ for Hsp90 was further supported by NMR spectroscopy of PTau$^{Cdk2}$ displaying only minor signal perturbations upon Hsp90 addition when compared to unmodified Tau (Fig. 6d), or compared to PTau$^{Cdk2}$ titrated with Hsp70:Hop:Hsp90 (Fig. 6e). This is also corroborated by ThT-based co-factor-free PTau$^{Cdk2}$ aggregation assays showing that PTau$^{Cdk2}$ fibrillization is inhibited

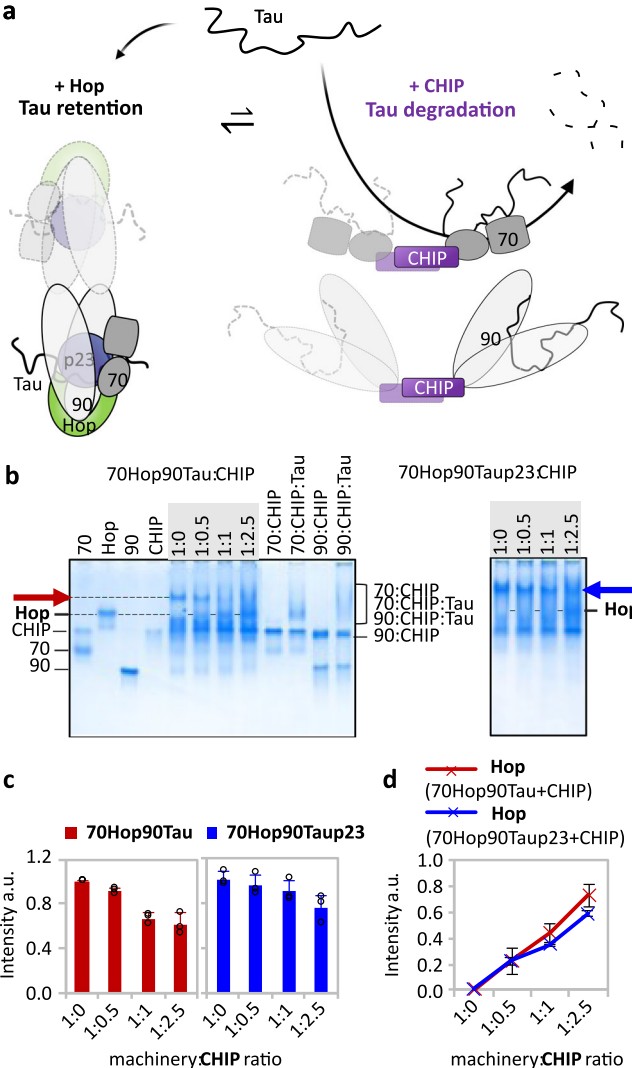

**Fig. 5 The E3-ubiquitin ligase CHIP disassembles the Hsp70/Hsp90 chaperone machinery. a** Schematic representation of the dynamic equilibrium between protein retention (via Hop) and protein degradation (via CHIP). CHIP is able to dissociate the Hsp70:Hop:Hsp90:Tau:p23 complex leading to Hsp70:CHIP:Tau and Hsp90:CHIP:Tau subcomplexes (purple). Note that the scheme portrays the possibilities of forming of 1:1 to 2:2 complexes of Tau:machinery, as suggested by SEC experiments. **b** Native PAGE analysis reveals that the dissociation of the Hsp70:Hop:Hsp90:Tau (red arrow) and Hsp70:Hop:Hsp90:Tau:p23 complex (blue arrow) results in Hsp70:CHIP, Hsp70:Tau:CHIP, Hsp90:CHIP and Hsp90:Tau:CHIP subcomplexes. Chaperone:CHIP(:Tau) molar ratios of 1:1(:5) were used for control experiments. Hsp70 and Hsp90 are abbreviated as "70" and "90", respectively. **c** Quantitative analysis of the band intensities from **b** corresponding to the Hsp70:Hop:Hsp90:Tau (red) and Hsp70:Hop:Hsp90:Tau:p23 complex (blue) in the presence of increasing concentrations of CHIP. Bar plots are shown as mean ± SD from three independent experiments. **d** CHIP-induced release of Hop from the Hsp70:Hop:Hsp90:Tau (red) and the Hsp70:Hop:Hsp90:Tau:p23 complex (blue) monitored by native PAGE band intensities in **b**. Data are shown as mean ± SD from three independent experiments. Source data are provided as a source data file.

more effectively by Hsp70:Hop:Hsp90:p23 than by Hsp90 alone (Fig. 6f, g). Interestingly, Hsp90 alone accelerates PTau$^{Cdk2}$ fibrillization while keeping a modest inhibitory effect (Fig. 6f–h). The addition of p23 to the Hsp70:Hop:Hsp90 machinery is crucial

in enhancing the inhibitory effect on PTau$^{Cdk2}$ fibrillization (Fig. 6f, g), but only slightly delays fibril formation (Fig. 6h).

Under our heparin-free aggregation conditions, both Tau and PTau$^{CDK2}$ aggregate in a sigmoidal fashion in the absence of chaperone machinery components (Figs. 2e, 6f). Noticeably, for PTau$^{CDK2}$ the aggregation profile appears more complex in the presence of chaperones. Bell-shaped ThT curves were observed for PTau$^{Cdk2}$:Hsp70:Hop:Hsp90 and PTau$^{CDK2}$:Hsp70:Hop:Hsp90:p23, and a wave-like pattern was shown for PTau$^{CDK2}$:Hsp90 (Fig. 6f). These unusual ThT curves were not observed for Tau with chaperone machinery components, which suggests that alternate aggregation pathways that resulted in amyloid-deficient PTau$^{CDK2}$ assemblies were promoted by the interaction between PTau$^{CDK2}$ and the chaperone machinery components. Under these conditions it is possible that PTau$^{CDK2}$ formed ThT-negative oligomeric aggregates or fibril polymorphs that led to the fluctuations in ThT intensity.

Overall, the finding that PTau$^{Cdk2}$ efficiently interacts with the Hsp70/Hsp90 multichaperone complex, but not with Hsp90 alone (Fig. 6), suggests that the assembly of Hsp90 into the Hsp70/Hsp90 multichaperone complex is important for Tau retention. Hsp90-mediated chaperoning pathways—including Tau degradation induced by the Hsp90/Tau/CHIP interaction (Fig. 5)—might be no longer accessible, potentially prohibiting or delaying the degradation of pathologically modified Tau during disease.

## Discussion

The combined data establish the Hsp70/Hsp90 multichaperone complex as a critical regulator of Tau homeostasis in neurodegenerative disorders. We reveal that the Hsp70/Hsp90 machinery has a critical holding function for Tau and likely other intrinsically disordered proteins. This holding function is specific for the Hsp70/Hsp90 machinery and does neither depend on energy consumption nor the interaction with nucleotides.

A key finding of our study is that pathologically phosphorylated Tau is efficiently chaperoned by the Hsp70/Hsp90 machinery, but can only weakly bind to Hsp90 alone. Previously, Hsp90 inhibition as well as p23 silencing have been described to decrease the levels of phosphorylated Tau in vivo[35], a finding that can now be ascribed to phosphorylated Tau molecules incorporated into the p23-stabilized Hsp70/Hsp90 multichaperone complex. Thus, despite its intended protective role to inhibit Tau aggregation, the Hsp70/Hsp90 chaperone machinery might simultaneously bear a harmful holding function, thereby fatally increasing the levels of pathologically modified Tau in vivo.

In contrast to previous findings suggesting that Hop and p23 would not be present together in one complex[36,37], we found that p23 associates with the Hsp70:Hop:Hsp90 complex in the presence of Tau. In addition, we observed enhanced/more stable binding of Tau to the multichaperone complex when p23 was present. Lower amounts of unbound Hsp70:Hop:Hsp90 machinery upon addition of p23 and Tau versus Tau alone were detected by native page and SEC (Figs. 2b, 4e). In addition, higher CHIP concentrations were required to dissociate comparable amounts of the Hsp70:Hop:Hsp90:Tau:p23 when compared to the Hsp70:Hop:Hsp90:Tau complex (Fig. 5b). The combined data indicate that p23 strengthens the binding of Tau to the multichaperone complex.

So far, the Hsp70/Hsp90 multichaperone machinery is suggested as a promoter of protein folding and activation[31]. For globular proteins, the chaperones are required to obtain distinct nucleotide states for client processing[36,38,39]. Our finding that nucleotides are not required for Tau chaperoning is however in agreement with previous results that, for the substrate Tau, the

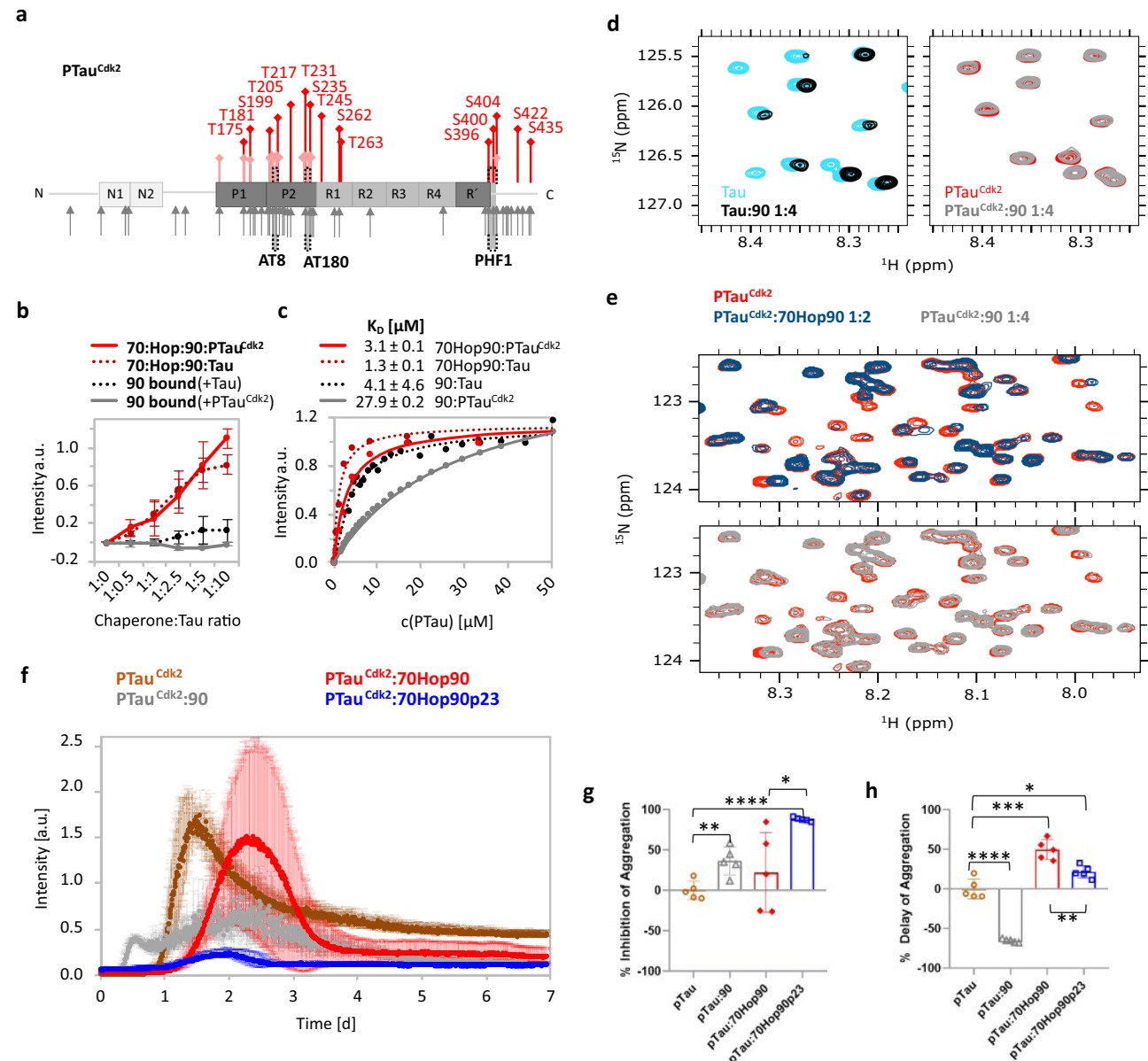

**Fig. 6 Pathologically modified Tau binds to the Hsp70/Hsp90 multichaperone chaperone complex but not to Hsp90 alone. a** Phosphorylation of Tau by Cdk2. Red bars mark Cdk2 phosphorylation sites determined by LC–MS/MS. Light red bars mark previously reported Cdk2 phosphorylation sites[66]. The main phosphorylation sites on Tau extracted from the brains of patients with Alzheimer's disease are indicated (gray arrows)[67]. Phospho-Tau antibody epitopes (AT8 (S202 and T205), AT180 (T231 and S235) and PHF1 (S396 and S404)) from paired helical filaments are highlighted with black brackets[34]. **b** Recognition of Cdk2-phosphorylated Tau by the Hsp70/Hsp90 multichaperone complex monitored by native PAGE. Displayed are band intensities derived from Supplementary Fig. 4. Increasing concentrations of PTau^Cdk2 push the equilibrium towards the Hsp70:Hop:Hsp90:PTau^Cdk2 complex (red solid line). In contrast, PTau^Cdk2 does not interact with Hsp90 alone (gray solid line). For comparison, the data obtained with unmodified Tau are shown in dashed lines. Data are shown as mean ± SD from three independent experiments. **c** Affinities of PTau^Cdk2 (solid line) and unmodified (dotted line; from Fig. 1d) Tau to the Hsp70/Hsp90 multichaperone complex (red), and to Hsp90 alone (gray/black). Errors represent SD from two independent experiments (70Hop90:PTau^Cdk2) or the standard error from the nonlinear fit (90:PTau^Cdk2). **d** 2D ^15N-^1H HSQC spectra of Tau (left panel) and PTau^Cdk2 (right panel) in the absence (cyan and red) and presence of Hsp90 (black and gray) using a Tau (or PTau^Cdk2):Hsp90 molar ratio of 1:4. **e** 2D ^15N-^1H HSQC spectra of PTau^Cdk2 (red) in the presence of the Hsp70:Hop:Hsp90 complex (top panel) and Hsp90 only (lower panel). Tau:chaperone molar ratios used are indicated on top of the spectra. **f** Aggregation assay of PTau^Cdk2 alone and in the presence of the Hsp70/Hsp90 chaperone machinery (Tau:machinery or Tau:chaperone molar ratios of 1:0.2). For panels **f-h**, data are presented as mean values ± SD from n = 5 independent experiments and null hypothesis testing was performed by using an unpaired, two-tailed t-test (*p ≤ 0.033, **p ≤ 0.0021, ***p ≤ 0.0002, ****p ≤ 0.0001). **g** Inhibition of PTau^Cdk2 aggregation from panel **f**. **h** Delay of PTau^Cdk2 aggregation from **f**. Source data are provided as a source data file.

affinity for Hsp90 remained unaffected with or without ATPγS[40]. It remains to be seen how the Tau-machinery interaction in the cell (with a pool of nucleotides in the cytosol) differs from what we observed in the absence of nucleotides in vitro. Moreover, we found that the Tau interaction with the Hsp70/Hsp90 chaperone machinery was independent of the co-chaperone Hsp40 (Supplementary Fig. 1d), which is known to assist the Hsp70:substrate interaction by stabilizing the chaperone's ADP-state[6]. Hence, with regard to Tau as an intrinsically disordered protein, the interaction with the Hsp70/Hsp90 chaperone machinery might be

because of protein holding rather than protein folding, which in turn does not necessarily require energy. Instead of the event of ATP hydrolysis that is reported to complete the Hsp90 action for other substrates[8], additional co-chaperones such as Aha1 or PPIases could, similarly to what has been shown with CHIP (Fig. 5), regulate the dynamic assembly and disassembly of the Hsp70/Hsp90 chaperone machinery:Tau complex.

The identification of the Hsp70/Hsp90 multichaperone complex as a critical regulator of Tau homeostasis is consistent with the importance of multichaperone assemblies for the regulation of protein homeostasis in neurodegeneration. Eukaryotic Hsp70 and Hsp90 are known to act through an orchestrated interplay, forming distinct machinery assemblies to accomplish diverse, though target-oriented activities[41]. Accordingly, the synergistic function of multiple chaperones in a disaggregation machinery is required for the disassembly of Tau amyloid fibrils[42]. Moreover, the Hsp70/Hsp90 multichaperone complex may also play a pivotal role in Parkinson's disease given that α-synuclein interacts with multiple chaperones including Hsp70 and Hsp90 in vitro and in mammalian cells[43].

The identification of the Tau:Hsp70/Hsp90 machinery complex provides novel therapeutic opportunities to counteract Tau aggregation and neurodegeneration. Indeed, a distinct Hsp90 conformation with high affinity for small molecules was observed in an Hsp70/Hsp90 multichaperone complex that drives tumor selectivity[12]. Further taking into account that pathologically modified Tau is a high-affinity substrate of the Hsp70/Hsp90 chaperone machinery, we suggest that targeting Hsp90 and other components of the Hsp70/Hsp90 multichaperone complex might be a powerful approach to specifically target the most toxic Tau species.

## Methods

**Protein production**. Full-length human DNA of Hsp40 (DNAJB4), Hsp70 (HSPA1A), Hop (STIP1), Hsp90 (HSP90AB1) and p23 (PTGES3) were cloned into the pET28a vector (Novagen) each with an N-terminal His₆-tag followed by a thrombin cleavage site for tag removal. pET28a-CHIP (STUB1) was received as a gift from Laura Blair[44].

Recombinant proteins were expressed in either Rosetta 2 (DE3) *Escherichia coli* cells (20 h at 20 °C for Hsp40, Hop and p23; 4 h at 37 °C for Hsp70 and Hsp90) or in BL21 (DE3) *E. coli* cells (20 h at 20 °C for CHIP) growing in LB medium. For NMR experiments, Hsp90 was produced perdeuterated with selectively labeled [$^1$H-$^{13}$C]-isoleucine δ-methyl groups by adding the metabolic precursor 2-ketobutyric acid-4 [$^{13}$C],3,3-[D₂] (NMR Bio SLAM Kit). Protein expression was performed in BL21 (DE3) *E. coli* cells growing in M9 minimal medium with 99% D₂O (20 h at 37 °C).

After cell lysis by sonication, His₆-tagged proteins were purified via Ni-NTA affinity chromatography (Qiagen). Respective binding buffers contained for Hsp40: 10 mM HEPES pH 7.5/500 mM NaCl/10 mM imidazole (500 mM for elution)/6 mM β-ME; for Hsp70: 20 mM Tris pH 8.0/500 mM NaCl/10 mM imidazole (250 mM for elution)/6 mM β-ME/1 mM RNase/1 mM DNase; for Hsp90: 20 mM Tris pH 8.0/500 mM NaCl/10 mM imidazole (250 mM for elution)/6 mM β-ME; for Hop and p23: 20 mM Tris pH 8.0/500 mM NaCl/10 mM imidazole (500 mM for elution)/3 mM β-ME; for CHIP: 20 mM Tris pH 8.0/500 mM NaCl/10 mM imidazole (500 mM for elution)/5 mM β-ME—each with freshly added 1 μM PMSF and 1 tablet of protease inhibitor cocktail III EDTA free (Millipore). Tags were cleaved off with thrombin protease in 20 mM Tris pH 8.0/150 mM NaCl/1 mM DTT (Hsp70) or 6 mM β-ME (p23 and CHIP). In the case of Hsp70, a second Ni-NTA affinity chromatography was performed to remove uncleaved fractions. Proteins were further purified via size exclusion chromatography using an SD200 (Hsp90) or SD75 column (Hsp40, Hsp70, Hop, p23, CHIP) (GE Healthcare). Respective SEC buffers contained for Hsp40, Hsp70 and CHIP: 25 mM HEPES pH 7.4/100 mM KCl/5 mM MgCl₂/1 mM TCEP; for Hop: 25 mM HEPES pH 7.4/150 mM NaCl/1 mM DTT; for Hsp90: 10 mM HEPES pH 7.5/500 mM KCl/0.5 mM DTT; for p23: 20 mM Tris pH 7.0/100 mM NaCl/1 mM DTT. All proteins were concentrated and flash frozen in 25 mM HEPES pH 7.4/100 mM KCl/5 mM MgCl₂/1 mM TCEP.

Full-length human Tau (MAPT) as well as its two shorter constructs K32 (P2-R') and K18 (R1–R4) were expressed in the pNG2 vector (Novagen) in BL21 (DE3) *E. coli* cells (4 h at 37 °C) growing in LB medium, or, to produce $^{15}$N-labeled Tau, growing in M9 minimal medium containing $^{15}$NH₄Cl as the only source of nitrogen. Tau, K32 and K18 were purified using a previously reported protocol, with a few modifications as described here[45]. The bacterial cells were harvested by centrifugation at 4000 × *g*, resuspended in lysis buffer (20 mM MES pH 6.8, 1 mM

EGTA, 2 mM DTT, 0.2 mM MgCl₂, 1 mg/mL lysozyme, 10 μg/mL DNAse I), and lysed using a French pressure cell. The resulting cell lysate was supplemented with NaCl to a final concentration for 500 mM NaCl, and boiled for 20 min to denature proteins. The lysate was gradually cooled to 4 °C, and ultracentrifuged (at 127,000 × *g*, 4 °C, 40 min). The supernatant was incubated with streptomycin (20 mg per 1 mL supernatant) for 15 min on ice to precipitate nucleic acids. The supernatant was pelleted by centrifugation (at 15,000 × *g*, 4 °C, 30 min). The cleared supernatant was incubated with 361 mg/mL (NH₄)₂SO₄ to precipitate Tau protein (also applies for K18, K32). The pellet collected by centrifugation (30 min, 4 °C, 15,000 × *g*) and resuspended in dialysis (20 mM MES pH 6.8, 1 mM EGTA, 2 mM DTT, 0.1 mM PMSF, 50 mM NaCl, 1 mM MgCl₂). The dialysate was ultracentrifuged for a second time (127,000 × *g*, 4 °C, 40 min) and the clarified supernatant was subjected to cation exchange chromatography (MonoS 10/100, 2 mL/min), in which the tau protein was eluted using a linear gradient of Buffer B (Buffer A is the same as the dialysis buffer, and buffer B is 20 mM MES pH 6.8, 1 M NaCl, 1 mM EGTA, 2 mM DTT, 1 mM MgCl₂, 0.1 mM PMSF). Fractions of tau protein eluting at ~60% buffer B were collected and concentrated to about 10 mg/mL protein. The protein concentrate was subjected to two rounds of gel filtration chromatography (Superdex 75 26/600, 2 mL/min) with gel filtration buffer (25 mM HEPES pH 7.4, 100 mM KCl, 5 mM MgCl₂, 1 mM TCEP). The eluted Tau protein fractions were analyzed by SDS-PAGE and fractions with >90% purity were combined and concentrated. Protein concentration was determined by bicinchoninic acid (BCA) assay (Pierce BCA Protein Assay Kit). Pure Tau/K18/K32 samples were flash-frozen in liquid nitrogen and stored at −80 °C until use in further experiments.

**In vitro complex reconstitution**. Complexes were formed in vitro in 25 mM HEPES pH 7.4/100 mM KCl/5 mM MgCl₂/1 mM TCEP by mixing the respective proteins in a ratio of 1:× (whereby 1 refers to 10 μM of protein solution, if not otherwise stated) in a final volume of 10 μL and shaking gently at 350 rpm for 45 min at 25 °C. For the initial tests, nucleotides (ATP, AMP-PNP, and ADP) were added in excess with 5 mM final concentration. After incubation, the protein solutions were mixed with native sample buffer (1:1; BioRad; 62.5 mM Tris pH 6.8/40 % (v/v) Glycerol/0.01 % (w/v) Bromophenol Blue) and analyzed via native PAGE using 7.5 % precast gels (Mini-Protean TGX; BioRad) running in 1 × TG buffer (BioRad; 25 mM Tris pH 8.3/192 mM Glycine) for 1.5 h at 110 V.

The band intensities $I_x$ (in a.u., arbitrary units) were quantified with ImageJ v1.52n and v1.53e and normalized using the equation $I_x = (I_m − I_{bg})/I_{ref}$, where $I_m$ is the mean intensity of the band of interest, $I_{bg}$ is the mean intensity of the background and $I_{ref}$ is the mean intensity of the reference band. Errors represent the standard deviation from three independent experiments.

**Affinity determination**. The affinity of Tau for the Hsp70/Hsp90 chaperone machinery was determined based on the complex band intensity obtained by native PAGE. The Hsp70:Hop:Hsp90 complex was mixed in a molar ratio of 1:1:1 (0.8 μM each) with and without 5 mM AMP-PNP and incubated with increasing amounts of Tau for 45 min at 25 °C gently shaking with 350 rpm. Samples were further analyzed by native PAGE as described above. The affinity of Tau for Hsp90 was redefined and determined by Trp fluorescence as described before[40]. In either case, the dissociation equilibrium constant $K_D$ was determined with the equation $I = I_{max} \cdot \frac{Tau_{total}}{Tau_{total} + K_D}$ using the solver function in Excel v16.43. This equation is based on the simplified assumption that the binding stoichiometry of Tau to the chaperone machinery is 1:1. Affinity measurements with unmodified and Cdk2-phosphorylated Tau were performed similarly. Errors represent the standard deviation from three independent experiments (70:Hop:90:Tau), the standard deviation of the fit (70:Hop:90:Tau (AMP-PNP), 90:Tau and 70:Hop:90:PTau$^{Cdk2}$) or the averaged standard deviation from two independent experiments (90:PTau$^{Cdk2}$).

**NMR spectroscopy**. NMR experiments were acquired on a Bruker Avance 900 MHz spectrometer equipped with a TCI cryogenic probe. Two-dimensional $^1$H-$^{15}$N HSQC NMR spectra were recorded at 5 °C with 24 μM of $^{15}$N-$^1$H labeled Tau in 25 mM HEPES pH 7.4/100 mM KCl/5 mM MgCl₂/1 mM TCEP/5% D₂O/0.02% NaN₃ in the absence and presence of Hsp70:Hop:Hsp90 (Tau:Hsp70HopHsp90 1:0.2, 1:1, 1:2) and p23 (Tau:Hsp70HopHsp90:p23 1:0.2:1 and 1:2:10) using 40 scans with 512 points in the indirect and 2048 points in the direct dimension. NMR experiments with unmodified and Cdk2-phosphorylated Tau were performed similarly.

Two-dimensional $^1$H-$^{13}$C methyl-TROSY data were collected at 25 °C with 50 μM of Hsp90 in the absence and presence of Hop (Hsp90:Hop 1:1), Hsp70 and Hop (Hsp70:Hop:Hsp90 1:1:1), likewise including Tau and p23 (Hsp70:Hop:Hsp90:Tau 1:1:1:5, Hsp70:Hop:Hsp90:Tau:p23 1:1:1:5:5), with all proteins dialyzed o/n into 25 mM HEPES pH 7.4/100 mM KCl/5 mM MgCl₂/1 mM TCEP in 99% D₂O. Each spectrum was recorded with 80 scans and 256 points in the indirect and 2048 points in the direct dimension[46].

Spectra were processed in TopSpin (Bruker) v3.6.1 and analyzed with Sparky (NMRFAM-SPARKY v1.4 powered by Sparky v3.13)[47]. Side chain specific chemical shift perturbations (CSPs) were calculated according to CSP$_{Tau}$ =

$\sqrt{\left(\left(\Delta\delta H\right)^2+\left(\frac{\Delta\delta N}{5}\right)^2\right)*\frac{1}{2}}$ and $\text{CSP}_{\text{Hsp90}}=\sqrt{\left(\left(\Delta\delta H\right)^2+\left(\frac{\Delta\delta C}{7}\right)^2\right)*\frac{1}{2}}$, where $\Delta\delta H$, $\Delta\delta N$ and $\Delta\delta C$ are the changes in proton, nitrogen, and carbon chemical shifts [ppm], respectively. Intensity plots represent the ratio of $I/I_0$, where $I$ is the peak intensity of the titration point and $I_0$ are the corresponding peak intensities in the reference spectra[48]. Error bars were calculated from the signal-to-noise ratio of each spectrum.

**Aggregation assay**. We followed the Tau aggregation assay as described recently[23] in the presence and absence of the Hsp70/Hsp90 chaperone machinery. Therefore, 25 μM of Tau were incubated in a total volume of 100 μL 25 mM HEPES pH 7.4/ 20 mM KCl/5 mM MgCl$_2$/1 mM TCEP/0.01% NaN$_3$/50 μM Thioflavin T (Tau:- machinery or Tau:chaperone molar ratio of 1:0.2) and 25 mM HEPES pH 7.4/ 35 mM KCl/5 mM MgCl$_2$/1 mM TCEP/0.01% NaN$_3$/50 μM Thioflavin T (Tau:- machinery molar ratio of 1:0.5). An Hsp70:Hop:Hsp90 and Hsp70:Hop:Hsp90:p23 molar ratio of 1:1:1 and 1:1:1:5 was applied, respectively. Each protein was mea- sured individually as a negative control using identical concentrations. ThT fluorescence was measured at 37 °C in a Tecan spark plate reader using the Spark control software v2.2 by Tecan every 10 min over a total time period of 1 week with λ$_{\text{excitation}}$ = 430 nm and λ$_{\text{emission}}$ = 485 nm. The samples were mixed throughout the experiment including each 3 polytetrafluoroethylene beads with a shaking frequency of 54 rpm. Error bars represent the standard deviation out of a minimum of three independent experiments. The % inhibition of Tau aggregation was cal- culated based on the maximum fluorescence intensity measured in each aggrega- tion experiment. To determine the % delay of Tau aggregation, ThT curves were fitted to a sigmoidal function and the time required to reach half-maximal fluor- escence (i.e. the midpoint of the growth phase of Tau fibril formation) was quantified for each sigmoidal curve using Graphpad Prism v8.0.1. Significant dif- ferences in % inhibition and % delay were determined by the unpaired two-tailed t- test in Graphpad Prism v8.0.1.

**Chemical crosslinking and mass spectrometry**. The Hsp70:Hop:Hsp90:Tau:p23 complex was mixed in a ratio of 1:1:1:5:5 in a final volume of 300 μL (35 μM chaperone concentration) and crosslinked with 1 mM disuccinimidyl suberate (DSS; ThermoFisher Scientific; stock solubilized in DMSO) or 10 mM 1-ethyl-3-(3- dimehtylaminopropyl) (EDC; with 25 mM Sulfo-NHS; ThermoFisher Scientific; stock solubilized in 25 mM HEPES pH 7.4/100 mM KCl/5 mM MgCl$_2$/1 mM TCEP) at 25 °C. The crosslinking reaction was quenched after 30 min (DSS) and 60 min (EDC) with 20 mM Tris pH 7.4. The crosslinked products were separated by sucrose gradient purification (50 μL sample per 4 mL sucrose gradient (10–25 %) in 25 mM HEPES pH 7.4/100 mM KCl/5 mM MgCl$_2$/1 mM TCEP) centrifuged for 16 h at 4 °C with 145,000 × g (SW 60 Ti rotor; Beckman Coulter). Fractions of 200 μL were taken top-down and protein contents were analyzed by SDS PAGE (4–15% gradient gel).

Fractions containing the complex were pooled from six gradients of the same kind. Pooled fractions were adjusted to 8 M urea, 50 mM NH$_4$HCO$_3$, 10 mM DTT and incubated 30 min at 37 °C. Proteins were alkylated in the presence of 40 mM iodoacetamide for another 30 min at 37 °C in the dark. The volume was adjusted to reach a final concentration of 1 M urea and 50 mM NH$_4$HCO$_3$. Aliquots of 650 μL of sample are mixed each with 50 μL SP3 beads and proteins are precipitated with 650 μL of 100 % ethanol in the presence of the beads, incubated for 5 min at room temperature, spun down and the beads were washed twice with 500 μL of 80 % (v/ v) ethanol. After complete removal of residual ethanol solution, beads were resuspended in 400 μL of 50 mM NH$_4$HCO$_3$, pH 8.0. Trypsin digest was performed overnight at 37 °C with 2.5 μg trypsin (Promega, V5111). Peptides were acidified with 4 μL of 100% formic acid (FA), desalted on MicroSpin columns (Harvard Apparatus) following manufacturer's instructions and vacuum-dried. Dried peptides were dissolved in 50 μL 30% acetonitrile/0.1% TFA and peptide size exclusion (pSEC, Superdex Peptide 3.2/300 column on an ÄKTAmicro system, GE Healthcare) was performed to enrich for crosslinked peptides at a flow rate of 50 μL/min. Fractions of 50 μL were collected. The first 8 or 9 fractions (after EDC or DSS crosslinking, respectively) containing crosslinked peptides were vacuum- dried and dissolved in 5% acetonitrile/0.05 % TFA (v/v) for analysis by liquid chromatography with tandem mass spectrometry (LC–MS/MS).

Crosslinked peptides derived from pSEC were analyzed as technical duplicates on a Q Exactive™ HF-X Hybrid Quadrupole-Orbitrap™ and Orbitrap Exploris 480 Mass Spectrometers (Thermo Scientific), coupled to a Dionex UltiMate 3000 UHPLC system (Thermo Scientific) equipped with an in-house-packed C18 column (ReproSil-Pur 120 C18-AQ, 1.9 μm pore size, 75 μm inner diameter, 30 cm length, Dr. Maisch GmbH). Samples were separated applying the following 58 min gradient: mobile phase A consisted of 0.1% formic acid (v/v), mobile phase B of 80% acetonitrile/0.08 % formic acid (v/v). The gradient started at 5% B, increasing to 10% B within 3 min, followed by 10–42% B within 35 min, 42–65% within 7.9 min, then increasing B within 0.6 min and keeping B constant at 90% for 4.5 min. After each gradient the column was again equilibrated to 5 % B for 7.5 min. The flow rate was set to 300 nL/min. MS1 spectra were acquired with a resolution of 120,000 in the Orbitrap covering a mass range of 350–2000 m/z. Injection time was set to 70 ms or 50 ms on Exactive™ HF-X Hybrid Quadrupole- Orbitrap™ and Orbitrap Exploris 480 Mass Spectrometer, respectively, and automatic gain control target to 3 × 10$^6$. Dynamic exclusion was set to 20 s. Only

precursors with a charge state of 3–8 were included. MS2 spectra were recorded with a resolution of 30,000 in the Orbitrap, injection time was set to 250 ms, automatic gain control target to 5 × 10$^5$ and the isolation window to 1.4 m/z. Fragmentation was enforced by higher-energy collisional dissociation at 30%. MS data were collected with Xcalibur v4.4.

For identification of crosslinked peptides, raw files were analyzed by pLink (v. 2.3.9), pFind group[49] using DSS or EDC as crosslinker and trypsin/P as digestion enzyme with maximal three missed cleavage sites. The search was conducted against a customized protein database containing all proteins within the complex. Carbamidomethylation of cysteines was set as a fixed modification, oxidation of methionines and acetylation at protein N-termini were set as a variable modifications. Searches were conducted in combinatorial mode with a precursor mass tolerance of 10 p.p.m. and a fragment ion mass tolerance of 20 p.p.m. The false discovery rate (FDR) was set to 0.01 (separate mode). Spectra of both technical duplicates were combined and evaluated manually. For further analysis only interaction sites with 3 crosslinked peptide spectrum matches were taken into account.

Crosslinked sites were plotted as networks with xiView[50] and mapped onto structures using Xlink Analyzer[51] combined with UCSF Chimera v1.14[52]. Intramolecular crosslinks were affirmed as valid according to the crosslinker length with a distance threshold set to 29.4 Å for DSS (=11.4 Å crosslinker arm + 2 × 6 Å for the two lysine side chains + 6 Å accounting for backbone flexibility) and 17 Å for EDC (=0 Å crosslinker arm + 6 Å for the two lysine side chain + 5 Å for the carboxylic acid side chain + 6 Å accounting for backbone flexibility)[53].

**Molecular weight determination**. After crosslinking with DSS and density gra- dient centrifugation, the complexes were analyzed by size exclusion chromato- graphy using an SD200 10 300 column (GE Healthcare) equilibrated in 25 mM HEPES pH 7.4/100 mM KCl/5 mM MgCl$_2$/1 mM TCEP. SEC chromatograms were collected with the UNICORN software v7.2. The molecular weight was determined according to the respective elution volume based on the reference curve $y = 5.1637x + 40.025$ (protein standard mix, 15–600 kDa; Sigma Aldrich). Error bars in the standard curve represent the peak widths at half height. Elution volumes of Hsp70:Hop:Hsp90, Hsp70:Hop:Hsp90:Tau and Hsp70:Hop:Hsp90:p23 with their corresponding calculated MW are reported as mean ± standard deviation from two independent experiments.

**Tau acetylation and phosphorylation**. Tau was acetylated by incubating the protein with 0.028 mg/mL of acetyltransferase (CBP or p300; Enzo Biochem Inc.) at a Tau:acetyl-CoA ratio of 10:1 in 25 mM HEPES pH 7.4/100 mM KCl/5 mM MgCl$_2$/1 mM TCEP/0.5 mM PMSF/5 mM EGTA. Tau phosphorylation was per- formed in 25 mM HEPES pH 7.4/100 mM KCl/5 mM MgCl$_2$/1 mM TCEP/1 mM PMSF/5 mM EGTA at a Tau:ATP ratio of 16:1 in presence of 0.02 mg/mL of the Cdk2/CyA2 kinase or 4 μmol/L of MARK2. Each reaction was incubated for 16 h at 30 °C gently shaking at 350 rpm. Enzymes were inactivated at 95 °C for 20 min and pelleted for 30 min at 4 °C with 14000 × g. The supernatant containing the mod- ified Tau was used for further experiments.

Subsequently, 20 μL of 20 μM Tau were used to determine the respective phosphorylation sites. To each sample, tris(2-carboxyethyl)phosphine and iodoacetamide in 25 mM of triethylamonium bicarbonat (TEAB) were added to final concentrations of 14 and 11.5 mM, respectively. After 1 h incubation at 37 °C in the dark, proteins were transferred to Amicon Ultra filters (0.5 mL; Merck) and buffers were exchanged to 8 M urea in 25 mM TEAB for four times and then to 25 mM TEAB for three times. Proteins were digested overnight in the filters with 0.5 μg trypsin at 37 °C. The resulting peptides were washed out from the filters two times with 200 μL of 0.1% formic acid (FA) and acidified with 10 μL of 10% FA. After C18 clean-up, the peptides were vacuum-dried and re-dissolved in 5% acetonitrile (ACN), 0.1% FA for LC–MS/MS analysis. Tryptic peptides were analyzed with a Q Exactive HF (HF; ThermoFisher Scientific) or an Orbitrap Fusion Lumos Tribrid (Lumos; ThermoFisher Scientific), both coupled with an UltiMate 3000 RSLCnano HPLC system (ThermoFisher Scientific). Peptides were first concentrated on a C18 pre-column (Dionex, 0.3 × 5 mm) and subsequently separated on a home-made analytical column (75 μm × 30 cm) packed with 1.9 μm ReproSil-Pur C18 AQ beads (Dr. Maisch GmbH) with a 1 h gradient.

The HF was operated in a top 20 data-dependent mode. MS1 scans were acquired with 120 K resolution, where the 20 most intense ions with charge 2–7 were selected for MS/MS with an isolation window of 1.4 m/z. The selected precursor ions were fragmented in HCD mode with normalized collision energy (NCE) of 28. MS2 fragment ions were acquired in the Orbitrap with a resolution of 30 K. The Lumas was operated in a top speed mode, where as many as the most abundant precursors were selected for MS/MS with a fixed three-second cycle time. Raw MS data were processed with MaxQuant (version 1.6.0.1)[54], and MS/MS spectra were searched against the human Tau protein sequence using the Andromeda search engine[55]. We set carbamidomethyl cysteine as a fixed modification. For variable modification, we used deamidated asparagine, oxidized methionine, and phosphorylated serine, threonine, and tyrosine. We enabled the LFQ and match between runs, while other settings maintained as default[56]. Phosphopeptides that were identified with 1% of false discovery rate (FDR) were further processed with Perseus to filter out decoy peptides and contaminants[57].

Phosphopeptide intensities were log2 transformed and normalized according to the intensity sum of all detected Tau peptides.

**Dynamic light scattering (DLS)**. Dynamic light scattering (DynaPro NanoStar; Wyatt) was used to determine the hydrodynamic radii of unmodified, acetylated and phosphorylated Tau using 10 μM of protein in a final volume of 20 μL 25 mM HEPES pH 7.4/100 mM KCl/5 mM $MgCl_2$/1 mM TCEP. Data were acquired and analyzed with DYNAMICS v7.10.0.23 (Wyatt Package). Errors represent the standard deviation from three independent measurements.

**Reproducibility and statistical analysis**. Each native PAGE and SDS-PAGE analysis was performed with $n = 3$ independent experiments showing similar results. Representative gels are shown in the figures and Supporting Information.

For aggregation assays, we used a minimum $n = 3$ (exact n values are provided in the figure legends) for all graphs and statistical tests. $P$ values were determined using the unpaired two-tailed $t$-test in Graphpad Prism v8.0.1. Differences were considered statistically significant when $p \leq 0.033$ (*$p \leq 0.033$, **$p \leq 0.0021$, ***$p \leq 0.0002$, ****$p \leq 0.0001$). Test statistics, $P$ values, degrees of freedom, effect sizes, and confidence intervals are provided in the source data files.

**Reporting summary**. Further information on research design is available in the Nature Research Reporting Summary linked to this article.

## Data availability

The data that support this study are available from the corresponding author upon reasonable request. All MS raw files were deposited to the ProteomeXchange Consortium (www.proteomexchange.org) via the PRIDE[58] partner repository with the dataset identifier PXD032037 [http://proteomecentral.proteomexchange.org/cgi/GetDataset?ID=PXD032037]. The cited PDB codes[59–63] 5fwk [https://doi.org/10.2210/pdb5FWK/pdb], 5aqz [https://doi.org/10.2210/pdb5AQZ/pdb], 4po2 [https://doi.org/10.2210/pdb4PO2/pdbU], 1elw [https://doi.org/10.2210/pdb1ELW/pdb], 1elr [https://doi.org/10.2210/pdb1ELR/pdb], 1ejf [https://doi.org/10.2210/pdb1EJF/pdb] are publicly available in the PDB. Source data are provided with this paper.

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

## Acknowledgements

We thank Laura Blair for providing the pET28a-CHIP DNA construct, Maria Sol Cima Omori for the production of the shorter Tau constructs K18 and K32, Filippo Favretto, Adriana Savastano and Gwladys Rivière for guidance during NMR experiments, Javier Oroz for useful discussions, Ralf Pflanz for initial crosslinking experiments, Kuan-Ting Pan for data analysis of the phosphorylation epitopes by LC–MS/MS, Monika Raabe for sample preparation and mass spectrometry analyses of crosslinked complexes, Erik Schliep for help with sucrose density gradient centrifugation, Ranjit Pradhan for support in PAGE experiments, and Pijush Chakraborty for the co-factor free aggregation assay protocol[23] and his help throughout the experiment and data analysis. H.U. is supported by the Deutsche Forschungsgemeinschaft, DFG (SFB1286/2, Project A10 and SFB860/3, Project A10). M.Z. was supported by the European Research Council (ERC) under the EU Horizon 2020 research and innovation program (grant agreement No. 787679).

## Author contributions

A.M. expressed and purified proteins, performed interaction studies, native PAGE, NMR and DLS experiments, affinity measurements, aggregation assays as well as the chemical crosslinking and data analysis, and molecular weight determinations. M.N. performed LC–MS/MS experiments and J.S. performed data analysis of the chemical crosslinked complexes. L.M.R. expressed and purified proteins and performed aggregation assays. H.U. supervised chemical crosslinking and LC–MS/MS studies. A.M. and M.Z. designed the project and wrote the paper.

## Funding

## Competing interests

The authors declare no competing interest.
