## [Peer Review File · Nature Communications]

Hsp multichaperone complex buffers pathologically modified
TauReviewers' Comments:

Reviewer #1:

Remarks to the Author:

The paper by Moll et al describes a multi-chaperone complex composed of Hsp70, Hsp90, Hop, and p23, that interacts with Tau through the aggregation-prone repeat region and, by doing so, prevents tau amyloid formation. The authors also interestingly find that, while Hsp90 chaperone by itself does not bind to hyperphosphorylated Tau, this aggregation-prone tau variant is recognized by the multi-chaperone complex.

The findings themselves are very interesting as the Hsp70-Hsp90 complexes are usually discussed in the context of misfolded proteins, yet their function on IPDs has not been described to date. Also the finding that only the chaperone complex can recognize the hyperphosphorylated form of Tau is of great importance, as tau hyperphosphorylation is known to be associated with neurodegenerative disorders.

There are however some points that must be addressed prior to publication -

1) The affinity of the multi-chaperone complex to Tau is calculated based on native gels band intensity, providing a relatively high affinity of 1.3 μ M. This affinity, however, is not consistent with the NMR measurements that show that even upon addition of 2-fold excess of the 710 kDa complex, \sim 70% of Tau peak intensities in the 15N-1H HSQC remain. Assuming a k_d of 1.3 μ M and the concentration used for NMR experiments (24 μ M Tau and 48 μ M of the multi-chaperone complex), over 90% of Tau should be bound. One would therefore expect significantly larger changes in Tau peak intensities, even without considering the relaxation contribution of forming a \sim 800 kDa bound complex

The fact that the majority of the Tau intensity was retained is particularly puzzling when considering that the Hsp90 methyl signals were broadened beyond detection upon the addition of Tau and p23 (Fig 4a).

2) In my opinion there isn't sufficient evidence presented in this paper to show that the multi-chaperone complex binds Tau better than the individual Hsp70 and Hsp90 chaperones, and the results in figure 2c even appear to show the opposite. There is a larger decrease in tau intensities upon addition of Hsp70 or Hsp90 chaperones alone compared to when the multi-chaperone complex is added (even though the complex is \sim 8 times larger than the chaperones). Perhaps aggregation-prevention experiments should be performed to demonstrate a synergistic (and not additive) effect of Hsp70 and Hsp90 as part of the multi-chaperone complex, compared to these chaperones alone.

3) The authors suggest that addition of p23 strengthens Tau binding to the Hsp70/Hsp90 chaperone machinery. The only evidence for this claim, though, is the observation that a slightly larger reduction in tau peak intensities is observed when the Hsp70/Hsp90/p23 complex is added compared to Hsp70/Hsp90 alone. Intensity measurements are, however, not an accurate method by which to determine the strength of an interaction, as they are also dependent on the increase in relaxation rates upon complex formation. Thus the slight changes observed could also result from the increase in the size of the complex.

Further experimental evidence would therefore be needed to support this claim. Should direct affinity measurements not be possible for this system, the authors can perform aggregation-prevention assays to show that lower Hsp70/Hop/Hsp90/p23 concentrations are required to prevent Tau aggregation compared to the same complex lacking p23

4) it is unclear how the conclusions regarding the 2:1:2 Hsp70:Hop:Hsp90 complex stoichiometry

were reached based on the methyl spectra. It is also not clear how the conclusion was reached that the complex with Hsp70 is not symmetrical. A more detailed explanation should be provided either in the text or in the methods section.

5) The results showing that the hyperphosphorylated Tau binds only to the multi-chaperone complex but not to the individual chaperones are very exciting and important for future drug development. This in my opinion is the major contribution of this manuscript. As such, a more rigorous analysis is required.

For example, binding of the multi-chaperone complex to Tau should also be demonstrated by NMR experiments. In addition, the affinity quantification should not be based solely on the native gels, as this method does not provide sufficiently accurate measurements.

Aggregation prevention assays could be another way to show the different effects of Hsp90 vs the multi-chaperone complex. In such assays, one would expect Hsp90 to have a minimal effect on pTau aggregation, while the complex should inhibit pTau and non phosphorylated Tau aggregation in a comparable manner.

Minor comments

1) In the native gels, Hsp70 is shown to migrate as an oligomer. This "oligo" band, however, does not change in intensity regardless of the components added. This is unexpected, as Hsp70 oligomers usually disassemble upon interaction with substrates or other chaperones. Can the authors perhaps elaborate more on why that would be?

2) In the native gels, the band for Tau alone is not visible. Could the authors provide an explanation as to why that is? A good control could perhaps be a western blot to ensure that Tau is indeed present in the multi-chaperone complex.

3) It seems that Tau binds Hsp90 in an additional region that is not involved in Hsp70 binding. This is mapped to the P2 region, which appears to be where p23 also binds Tau. This should perhaps be mentioned in the text.

4) In figure 3c, Hsp90 shows some binding to Hsp70 (without Hop). Was such an interaction previously described for human chaperones?

5) The use of a density gradient followed by SEC to get an exact mass is not reliable. The authors state that, in the presence of Tau, the complex stoichiometry changes and a tetrameric Hsp90 is formed while one Hsp70 is released from the original dimer. This conclusion cannot be reliably reached based on SEC alone. Perhaps SEC-MALS experiments with fluorescently-tagged Hsp70 should be performed instead.

6) According to the cross linking / MS data, all regions of Tau protein are involved in binding to both p23 and Hsp70. The NMR experiments, however, show very localized interaction, mainly in the repeat region. Could the authors elaborate on these discrepancies?

7) The authors should show the mass spec results of the hyperphosphorylated tau protein.

8) In figure 4 and supp Fig 3d - please put domain names for Tau protein as well.

9) There is no reference in the text to panel 5d.

10) In Figure 5b, please indicate what the red and blue arrows represent. Additionally, I did not find in the legend or methods what ratio of CHIP was added to the fractions in the right lanes, showing the

disassembly into 70:CHIP, 90:CHIP, etc.

Reviewer #2:

Remarks to the Author:

This is an interesting manuscript, which explores the biophysical interactions of tau with the Hsp70-HOP-Hsp90 complex. This chaperone complex has been best associated with folding and stabilization of steroid hormone receptors and kinases. In addition, the individual factors Hsp70 and Hsp90 have been extensively studied for their effects on tau aggregation and microtubule binding. In brief, the authors have combined those two ideas here – using NMR and X-linking MS to study the binding of tau to the Hsp70-HOP-Hsp90 complex. Two of the most important findings are that p23 promotes tight binding, at least partially by binding directly to tau, and the ability of CHIP to displace the complex. All together, this is an important work, as it extends previous studies on the individual, binary interactions and begins to probe the impact of larger systems (likely to increasingly simulate the endogenous machines).

1. It is a little surprising that nucleotide is not required for these interactions, as previous studies using the GR system have shown that ATP hydrolysis is involved in assembly and progression of the Hsp70-HOP-Hsp90-p23 complex (Kirshke 2014 Cell; Noddings 2021 BioRxv). The author's data is pretty clear, but this stark contrast with previous systems should be expanded upon in the discussion to make it clear that the tau system seems to have distinct features.
2. The lack of nucleotide dependence also begs the question of how the complex is dis-assembled (and, therefore, regulated)? The effects of CHIP are interesting, but even this factor was poorly able to act on the p23-containing complex. One wonders if nucleotide exchange factors, such as BAG3 (Kirshke 2014), can reverse the complex assembly – as seen in the GR and kinase complexes?
3. Similarly, another Hsp90 co-chaperone, Aha1, has been especially linked to tau proteostasis and the progression to aggregation (Shelton et al. 2017 PNAS; Criado-Marrero et al. 2021 Acta Neuropath. Commun; Singh et al. 2020 Cell Chem Biol.). If feasible, studying the effects of Aha1 on the dissolution of the Hsp70-HOP-Hsp90-tau complex would be interesting to add. Please note that this comment and the previous one are acknowledged to be a substantial amount of work.
4. The co-factor free tau aggregation method is well-used here, but it is not clear if the polytetrafluoroethylene beads used to induce aggregation might similarly aggregate chaperones? This seems like a critical control.
5. The ability of CHIP to displace the Hsp70-HOP-Hsp90-tau complex is interesting, but also somewhat unexpected. The affinity of CHIP and HOP for the EEVD motifs of Hsp70 and Hsp90 are (roughly) similar; so CHIP's ability to drive such complete dissolution of the complex at 1:1 stoichiometry is unexpected given the avidity expected within the pre-existing, large complex. What is the kinetics of this effect? One would expect it to be slow, as dissociation of the complex would seem to be required. An alternative is that CHIP binds directly to tau (Kim et al. 2021 Chem Sci 12:5599), so maybe this interaction also serves to assist in dissolving the complex?

Reviewer #3:

Remarks to the Author:

Hsp multichaperone complex buffers pathologically modified Tau

The details of interaction between Tau with Hsp90 have been investigated by combining NMR approach with small angle X-ray scattering (SAXS) and described in the author Zweckstetter's publication (Cell 2014, 27; 156(5): 963-74). In this reviewing paper, the authors observed the complex Hsp70:Hop:Hsp90:Tau and Hsp70:Hop:Hsp90:Tau:p23 formed on the native gels and interaction between proteins were detected by 900 MHz spectrometer NMR and crosslinking mass spectrometry. Overall, the results are compelling to support the authors' major conclusions. I have the following points about the manuscript.

1. The iBAQ method cannot provide a protein precise ratio of 1:1:1:1. The figure S1b shows the four proteins were identified in the complex of Hsp70:Hop:Hsp90:Tau. It is better to list the iBAQ intensity value obtained from the MaxQuant protein table than to show it on a figure

2. The stoichiometry of the Tau complex with the other three proteins in $2\text{Hsp70}:1\text{Hop}:2\text{Hsp90}$ and $(1\text{Hsp70}:1\text{Hop}:2\text{Hsp90}:1\text{Tau})_2$ complex has been defined by the density gradient centrifugation followed by size exclusion chromatography.

$2\text{Hsp70}:1\text{Hop}:2\text{Hsp90}$: 380kd (415kd by sequence calculation)

$(1\text{Hsp70}:1\text{Hop}:2\text{Hsp90}:1\text{Tau})_2$: 853kd (739kd by sequence calculation)

There is a large mass difference between theoretical calculation and measurement for $(\text{Hsp70}_1:\text{Hop}_1:\text{Hsp90}_2:\text{Tau}_1)_2$ even considering Tau's apparent mass different with actual mass (Ref. Acta Neuropathol 2017, 133:665–704). May it be a mixture of various ratio species? Does the measurement method have limitation? Can the authors make an explanation?

3. The complex assembly in Fig 4h are derived from the EM results of references [1-3] without any other modelling experiment results support e. g. crosslinking data with homology model from EM. Crosslinking mass spectrometry data have no contribution to solving this complex assemblies.

4. Questions following the Crosslinking experimental data

- a. This is a vitro crosslinking experiment in an aqua-buffer. The authors use a non-water soluble crosslinker DSS instead of the water soluble partner BS3. Why?
- b. Either DSS/BS3 or EDC may not be a good choice for unstructured Tau. NHS ester crosslinkers capture a lot of flexibility interaction (protein kinetics trapping). Photo-crosslinker e. g SDA is a better choice for detecting core interactions (J Proteome Res, 18 (2019), pp. 934-946).
- c. Peptide size exclusion (pSEC, Superdex Peptide 3.2/300 column on an ÄKTAmicro system, GE Healthcare) was performed to enrich for crosslinked peptides. The raw data showed crosslinking peptides distributed cross at least 10 fractions and have not been enriched properly

(Normally crosslinking peptides are enriched at 3-4 fractions on this column)

- d. The crosslinked samples for both DSS and EDC were overloaded and separated at a very short LC gradient (46.5min gradient in a 58 minute run) on LC-MS. There are a lot of peptides co-elution without proper separation and that has caused the crosslinks identified all over the sequences to contain a high percentage of false positives. The crosslinking data shown in Figure S4c are a part of crosslinks by 5% FDR plus high score cut off (which have not shown all the crosslinks that pass 5% FDR threshold). I identified around 2500 links with 1% FDR for duplications of EDC data sets with a setting of MS1 error 3 ppm and a MS2 error of 10 ppm in combined fraction 25-30).

- e. The crosslinking data (selected by 5% FDR then extra score cut off with at least three times of identification) were validated by five PDBs of Hsp90, Hsp70-NBD, Hsp70-SBD, Hop-TPR1, Hop-TPR2A, and p23. Only 64 % of the selected EDC links are within the distance constraints. Could the authors discuss why the percentage is so low?
- f. The crosslinked products were separated by sucrose gradient purification. Do the authors have any SDS gel image of crosslinked product bands? Is it a mixture of different complexes and sub-complexes?

5. The data deposit reference PXD024457 is the wrong identifier, the correct one is PXD027287

EM references

1. Wang, R. Y.-R., Noddings C. M., Kirschke E., Myasnikov A. G., Johnson J. L., and Agard D.A. GR chaperone cycle mechanism revealed by cryo-EM: inactivation of GR by GR:Hsp90:Hsp70:Hop client-loading complex. *bioRxiv*, 2020.2011.2005.370247, doi:10.1101/2020.11.05.370247 (2020).
2. Liu, Y., Sun, M., Elnatan, D., Larson, A. G., Agard, D. A. Cryo-EM analysis of human mitochondrial Hsp90 in multiple tetrameric states. *bioRxiv*, 2020.2011.2004.368837, doi:10.1101/2020.11.04.368837 (2020).
3. Liu, Y., Elnatan, D., Sun, M., Myasnikov, A. G. Agard, D. A. Cryo-EM reveals the dynamic interplay between mitochondrial Hsp90 and SdhB folding intermediates. *bioRxiv*, 2020.2010.2006.327627, doi:10.1101/2020.10.06.327627 (2020).

We thank the three reviewers for their consistently positive assessment of our work and for their many helpful comments and suggestions for improvements. In the following, our response is typed in blue, newly added text in italics. We believe that, thanks to the reviewer input, the manuscript has greatly improved.

Reviewer #1:

The paper by Moll et al describes a multi-chaperone complex composed of Hsp70, Hsp90, Hop, and p23, that interacts with Tau through the aggregation-prone repeat region and, by doing so, prevents tau amyloid formation. The authors also interestingly find that, while Hsp90 chaperone by itself does not bind to hyperphosphorylated Tau, this aggregation-prone tau variant is recognized by the multi-chaperone complex. The findings themselves are very interesting as the Hsp70-Hsp90 complexes are usually discussed in the context of misfolded proteins, yet their function on IPDs has not been described to date. Also the finding that only the chaperone complex can recognize the hyperphosphorylated form of Tau is of great importance, as tau hyperphosphorylation is known to be associated with neurodegenerative disorders.

There are however some points that must be addressed prior to publication -

1) The affinity of the multi-chaperone complex to Tau is calculated based on native gels band intensity, providing a relatively high affinity of 1.3 μM . This affinity, however, is not consistent with the NMR measurements that show that even upon addition of 2-fold excess of the 710 kDa complex, ~70% of Tau peak intensities in the 15N-1H HSQC remain. Assuming a k_d of 1.3 μM and the concentration used for NMR experiments (24 μM Tau and 48 μM of the multi-chaperone complex), over 90% of Tau should be bound. One would therefore expect significantly larger changes in Tau peak intensities, even without considering the relaxation contribution of forming a ~800 kDa bound complex. The fact that the majority of the Tau intensity was retained is particularly puzzling when considering that the Hsp90 methyl signals were broadened beyond detection upon the addition of Tau and p23 (Fig 4a).

Reply: We thank the referee for raising this issue. Please note that ligand-induced changes in NMR spectra strongly depend on the exchange rate of the binding process, as well as the chemical shift differences between the free and bound conformations. The binding of Tau to Hsp90/machinery occurs on a time scale that is intermediate on the chemical shift time scale. For intermediate exchange with $k_{\text{ex}} \sim |\Delta\omega|$ (k_{ex} is defined as the rate constant of the chemical exchange between free and bound states, and $\Delta\omega$ is chemical shift difference between free and bound states; Teilum et al. doi.org/10.1002/pro.3105), the NMR signal intensities of P do not accurately represent the population of unbound P since lines are broadened by the exchange between free and bound states of P.

In order to judge an affinity value with respect to changes in the NMR spectrum, it is thus better to compare individual spectra of Tau in the presence of different ligands. Indeed we found that the lower affinity of Hsp90 to Tau ($K_D = 4.1 \mu\text{M}$; also reported by Karagoz et al. [doi:10.1016/j.cell.2014.01.037](https://doi.org/10.1016/j.cell.2014.01.037)) is represented by smaller changes in the NMR spectrum when compared to the Tau:machinery interaction ($K_D = 1.3 \mu\text{M}$). A particular striking example of the difficulty to directly associate ligand-induced changes in the NMR spectra of an intrinsically disordered protein with the affinity of the interaction was provided in Borgia, A., Borgia, M., Bugge, K. et al. <https://doi.org/10.1038/nature25762> (2018), demonstrating affinity values in the pmol range and only small perturbations in NMR spectra. Please also note that the strong line broadening observed for the Hsp90 methyl signals is already apparent before the addition of Tau and thus mainly based on the interaction with Hsp70 and Hop (seen in Fig.4b, orange line).

To enable a direct comparison of the NMR line broadening caused by Hsp90 and the machinery, we modified Fig. 2c and Supplementary Fig. 2f. In Fig. 2c we replaced the NMR interaction profiles of Tau:Hsp70 (1:4) and Tau:Hsp90 (1:4) with those of the 1:2 molar ratios so that a direct comparison with the Tau:machinery (1:2) interaction can be seen. In addition, we added “*Results using*

Tau:chaperone molar ratios of 1:2 are shown for all experiments.” in the respective figure legend. Accordingly, we replaced the respective NMR spectra in the Supplementary Fig. 2f.

2) In my opinion there isn't sufficient evidence presented in this paper to show that the multi-chaperone complex binds Tau better than the individual Hsp70 and Hsp90 chaperones, and the results in figure 2c even appear to show the opposite. There is a larger decrease in tau intensities upon addition of Hsp70 or Hsp90 chaperones alone compared to when the multi-chaperone complex is added (even though the complex is ~8 times larger than the chaperones). Perhaps aggregation-prevention experiments should be performed to demonstrate a synergistic (and not additive) effect of Hsp70 and Hsp90 as part of the multi-chaperone complex, compared to these chaperones alone.

Reply: We understand that this comment is directly related to point 1). We hope to satisfactorily convince the referee with the adjustments we have made described in point 1) evidently showing that the multi-chaperone complex binds Tau better than the individual Hsp70 and Hsp90 (see updated Fig. 2c). We also performed the suggested aggregation-prevention experiments (updated Fig. 2e-g), which show that 70Hop90 and 70Hop90p23 both more effectively decrease Tau aggregate formation when compared to the individual Hsp70 and Hsp90 chaperones.

3) The authors suggest that addition of p23 strengthens Tau binding to the Hsp70/Hsp90 chaperone machinery. The only evidence for this claim, though, is the observation that a slightly larger reduction in tau peak intensities is observed when the Hsp70/Hsp90/p23 complex is added compared to Hsp70/Hsp90 alone. Intensity measurements are, however, not an accurate method by which to determine the strength of an interaction, as they are also dependent on the increase in relaxation rates upon complex formation. Thus the slight changes observed could also result from the increase in the size of the complex. Further experimental evidence would therefore be needed to support this claim. Should direct affinity measurements not be possible for this system, the authors can perform aggregation-prevention assays to show that lower Hsp70/Hop/Hsp90/p23 concentrations are required to prevent Tau aggregation compared to the same complex lacking p23.

Reply: We thank the referee for encouraging us to clarify this issue. The statement that p23 strengthens the binding of Tau to the multi-chaperone complex is based on four complementary observations:

- the binding behavior seen by native page (see Fig. 2b – less unbound 70Hop90 machinery base in the presence of p23 and Tau versus Tau only)
- the SEC elution profile (see Fig. 4e – the shoulder peak referring to the 70Hop90 machinery base is absent in presence of p23 and Tau versus Tau only)
- the discrete band of the DSS-crosslinked Hsp70:Hsp90:Tau:p23 complex observed in SDS-PAGE (please see updated Supplementary Fig. 3b) which is not seen in the absence of p23.
- the dissociation experiment with CHIP (see Fig. 5b – higher concentrations of CHIP were necessary to dissociate equivalent amounts of the 70Hop90Tau:p23 versus 70Hop90Tau complex).

To better reflect these results, we have added a separate paragraph to the discussion section of the revised manuscript (page 14): *“In contrast to previous findings suggesting that Hop and p23 would not be present together in one complex,^{36,37} we found that p23 associates with the Hsp70:Hsp90 complex in the presence of Tau. In addition, we observed enhanced/more stable binding of Tau to the multichaperone complex when p23 was present. Lower amounts of unbound Hsp70:Hsp90 machinery upon addition of p23 and Tau versus Tau alone were detected by native page and SEC (Fig. 2b and Fig. 4e). In addition, higher CHIP concentrations were required to dissociate comparable amounts of the Hsp70:Hsp90:Tau:p23 when compared to the Hsp70:Hsp90:Tau complex*

(Fig. 5b). The combined data indicate that p23 strengthens the binding of Tau to the multichaperone complex.”

We also performed aggregation-prevention assays of Tau with the Hsp70:Hop:Hsp90 system (with and without p23; Supplementary Figure 2g-h). We found that in the presence of p23 the inhibition is slightly less effective (for both mole ratios of 1:0.2 and 1:0.5 Tau:machinery). Nevertheless, when the concentration of the machinery is higher (e.g. 1:0.5), the extent of aggregation inhibition in the presence and absence of p23 is almost equal.

4) It is unclear how the conclusions regarding the 2:1:2 Hsp70:Hop:Hsp90 complex stoichiometry were reached based on the methyl spectra. It is also not clear how the conclusion was reached that the complex with Hsp70 is not symmetrical. A more detailed explanation should be provided either in the text or in the methods section.

Reply: We thank the referee for pointing this out; we agree that the data do not exclude the possibility of having a complex in which Hsp70 is not symmetrical. We therefore modified our description (page 8 of the revised manuscript): “Our observation of signal broadening (far below 50%) for methyl protons along the length of Hsp90 (Fig. 3b) upon titration with Hop and Hsp70 indicate that the Hsp90 domains are potential binding sites of Hsp70 while Hsp90 is associated with Hop. This may be in agreement with an assembly in which a single Hsp70 molecule binds Hsp90:Hop via Hop,²⁶ or one in which a single Hsp70 molecule is bound to each of the two Hsp90 molecules in the dimer as observed in a recent cryo-EM structure of a client-loading complex containing a Hsp70₂:Hop₁:Hsp90₂ machinery.²¹ Notably, in the same cryo-EM study, a stoichiometry of Hsp70₁:Hop₁:Hsp90₂ was also observed.²¹ Please note: we previously observed a 2:1 molar ratio of the Hsp90:Hop interaction (Lott, A, et al. <https://doi.org/10.1002/pro.3969> (2020)).

5) The results showing that the hyperphosphorylated Tau binds only to the multi-chaperone complex but not to the individual chaperones are very exciting and important for future drug development. This in my opinion is the major contribution of this manuscript. As such, a more rigorous analysis is required. For example, binding of the multi-chaperone complex to Tau should also be demonstrated by NMR experiments.

Reply: Thanks for this suggestion. In Fig. 6e we now show the NMR data for the interaction of PTau with the Hsp70:Hop:Hsp90 complex. The respective zoom-in of PTau with Hsp90 alone is shown for comparison within the same figure – using even double amount of Hsp90 compared to the multichaperone complex, at best very weak interaction is observed for Hsp90 alone.

In addition, the affinity quantification should not be based solely on the native gels, as this method does not provide sufficiently accurate measurements.

Reply: We are absolutely compliant with the concerns of the referee that native page would not be the first method of choice for affinity quantification. However, according to our experience working with hyperphosphorylated Tau native page turned out to be the most reliable technique for affinity quantification with respect to the machinery. Prior to the initial submission of this manuscript we additionally performed numerous attempts of Trp fluorescence quenching as well as ITC experiments, which both failed to give reliable outcomes. We attribute this to the complexity of the multichaperone system, to which hyperphosphorylated Tau further contributes to increased heterogeneity of the sample (see also Supplementary Fig. 4b): in Trp fluorescence experiments we believe that the inconsistent results arise from the fact that upon the interaction, the Trp fluorescence of one protein might be quenched whereas the fluorescence of another protein might be enhanced so that the effects eventually ambiguously overlap. The same might hold true for the ITC experiments with regard to

opposing enthalpic effects (exothermic and endothermic). We are thus convinced that the native page analysis currently provides the most reliable affinity quantification for the pTau/machinery interaction.

Aggregation prevention assays could be another way to show the different effects of Hsp90 vs the multi-chaperone complex. In such assays, one would expect Hsp90 to have a minimal effect on pTau aggregation, while the complex should inhibit pTau and non-phosphorylated Tau aggregation in a comparable manner.

Reply: We performed the suggested aggregation assays on pTau (phosphorylated by Cdk2; results are shown Figures 6f-h). We observed that Hsp90 overall decreases pTau aggregate formation (quantified by the maximal Thioflavin-T fluorescence intensity), but results in faster aggregation kinetics (judging from the time it took to reach half-maximal fluorescence value in the course of the aggregation assay). This was not the case observed for Tau aggregation in the presence of Hsp90 (Fig. 2e,f), and hence reflects the different affinities of Hsp90 for Tau vs pTau. In addition, the inhibition of aggregation is more effective for both Tau (Figure 2e,f) and pTau (Figure 6f,g) in the presence of the Hsp70:Hsp90:p23 complex than in the presence of Hsp90 alone.

Minor comments

1) In the native gels, Hsp70 is shown to migrate as an oligomer. This “oligo” band, however, does not change in intensity regardless of the components added. This is unexpected, as Hsp70 oligomers usually disassemble upon interaction with substrates or other chaperones. Can the authors perhaps elaborate more on why that would be?

Reply: In general we also were expecting that the “Hsp70” oligo band would change, for example upon addition of the chaperones. This however was not the case, suggesting that this is sort of a pool of “inactive” Hsp70 oligomers. Because we currently do not know the origin/properties of these species, we feel that we are currently not in a position to provide further insights about this band.

2) In the native gels, the band for Tau alone is not visible. Could the authors provide an explanation as to why that is? A good control could perhaps be a western blot to ensure that Tau is indeed present in the multi-chaperone complex.

Reply: The native page experiments were performed in the TG running buffer from BioRad having a pH of 8.3. With a theoretical isoelectric point of $pI = 8.24$ the protein Tau has too low charge to be able to run into the gel effectively and thus only a small amount of Tau enters the gel in its unbound state. For more clarity, we have now included the components of the BioRad solutions used in the methods section (to be found on p. 17). We however verified the presence of all components of the Hsp70/Hsp90 chaperone machinery: Tau multichaperone complex using MS (please see Supplementary Table 1).

In addition we now performed a native page experiment followed by Western Blotting as suggested by the referee. Two native page gels were run in parallel (same orientation as the page in Fig. 1b), one imaged with Coomassie Blue staining (Panel A) and the other was subjected to typical Western Blot protocols, e.g. a semi-dry transfer to a nitrocellulose membrane, followed by binding of anti-Tau primary antibody and a secondary antibody (Panel B, Western Blot). The Western Blot shows that a smaller amount of unbound Tau enters the gel ("Tau" Lane) compared to the bound condition in which the same total concentration of Tau is incubated together with the 70Hop90 machinery ("1:1" Lane). Moreover, the Western Blot confirms the presence of high molecular weight bands containing Tau in the conditions 1:1 to 1:10 (70Hop90: Tau). This confirms that Tau is present in the multichaperone complex.

3) It seems that Tau binds Hsp90 in an additional region that is not involved in Hsp70 binding. This is mapped to the P2 region, which appears to be where p23 also binds Tau. This should perhaps be mentioned in the text.

Reply: We thank the referee for pointing this out. We included this observation in the results section (page 9): *"In agreement with a direct Tau:p23 interaction demonstrated by NMR (Fig. 2d), the majority of p23 cross-links were found with the P2 region of Tau (Fig. 4c, d and Supplementary Fig. 3e). Notably, the P2 region was shown to be the additional binding site for Hsp90 that is not involved in Hsp70 binding (Fig. 2c)."*

4) In figure 3c, Hsp90 shows some binding to Hsp70 (without Hop). Was such an interaction previously described for human chaperones?

Reply: Indeed a direct (maybe weak) interaction between Hsp70 and Hsp90 was detected by NMR in the absence of Hop (Fig. 3a, b). We are not aware that this has been reported before. Please note that the observed Hsp70 binding site in Hsp90's NTD-MD region in the presence of Hop (Fig. 3c) overlaps with the one reported for the bacterial/yeast proteins in the absence of Hop (Genest et al. doi:10.1016/j.jmb.2015.10.010 (2015); Doyle S. et al. doi:10.1016/j.jmb.2019.05.026 (2019)) indicating a conserved Hsp70:Hsp90 interaction site.

5) The use of a density gradient followed by SEC to get an exact mass is not reliable. The authors state that, in the presence of Tau, the complex stoichiometry changes and a tetrameric Hsp90 is

formed while one Hsp70 is released from the original dimer. This conclusion cannot be reliably reached based on SEC alone. Perhaps SEC-MALS experiments with fluorescently-tagged Hsp70 should be performed instead.

Reply: We agree with the reviewer that the mass determined by SEC is merely an estimation. We therefore revised the text as follows (page 10): *“For the Hsp70:Hop:Hsp90 complex in the absence of Tau, we obtained an elution volume corresponding to a molecular mass of 411 kDa (Fig. 4e, f). In the presence of Tau, however, the estimated mass increased to 739 kDa (Fig. 4e, f), i.e. much larger than the molecular weights expected for a Hsp70₂:Hop₁:Hsp90₂:Tau₁ complex (425 kDa) and a Hsp70₁:Hop₁:Hsp90₂:Tau₁ complex (354 kDa). However, given the broad nature of the complex peak in SEC it cannot be excluded that multiple stoichiometries ranging from ~400 kDa to ~700 kDa may be formed (corresponding to elution volumes ~11 mL to ~9 mL). This is consistent with SDS-PAGE analysis of the cross-linked Hsp70:Hop:Hsp90:Tau complexes that show band smearing corresponding to molecular weights above 200 kDa (Supplementary Fig. 3b). The wide range of stoichiometries may correspond to 1:1 to 2:2 complexes of Tau:machinery, with possibly one or two Hsp70 units within the Hsp70:Hop:Hsp90 machinery. Notably, in the presence of p23 the main cross-linked complex shows a quite defined band in SDS-PAGE (Supplementary Fig. 3b) as well as a narrower peak in the SEC chromatogram (~700 kDa molecular weight corresponds to the elution volume at the peak maximum, Fig. 4e). This suggests that the addition of p23, which directly binds to Tau (Fig. 2d), further stabilized the high-molecular weight complex. The combined data point to the formation of high molecular complexes of Hsp70:Hop:Hsp90:Tau and Hsp70:Hop:Hsp90:Tau:p23 with variable stoichiometry.”*

6) According to the cross linking/MS data, all regions of Tau protein are involved in binding to both p23 and Hsp70. The NMR experiments, however, show very localized interaction, mainly in the repeat region. Could the authors elaborate on these discrepancies?

Reply: We believe that Fig. 4c clearly shows that most of the cross-links were found with the central part of Tau, regardless of the interaction partner. Nevertheless, it is true that these data show additional cross-links in the terminal regions of Tau with p23 and Hsp70. Please note that the detected cross-links were colored according to the count of crosslinked peptide spectrum matches (CSMs), so that the darker the color, the more reliable the crosslink. In this context, the lighter colored crosslinks between p23 and Tau, which in addition are scarce at the terminal regions of Tau, should be considered with caution and may be ascribed to a transient interaction. The same holds true for the interaction of Tau with Hsp70: again, only a few connections to the N-terminal region of Tau were detected and correspond to a low count (<6). Thus, these crosslinks are likely to arise from transient interactions, too. In general, one should keep in mind that, also considering the NMR data in Fig. 2c, the terminal regions and especially the N-terminal domain of Tau are suggested to remain highly flexible even in the multi-chaperone complex, therefore transient interactions within these parts are conceivable.

7) The authors should show the mass spec results of the hyperphosphorylated tau protein.

Reply: We included the MS data as Supplementary Table 4 (referred to in the results section; page 12).

8) In figure 4 and supp Fig 3d - please put domain names for Tau protein as well.

Reply: Done.

9) There is no reference in the text to panel 5d.

Reply: Thanks for pointing it out. It is now corrected.

10) In Figure 5b, please indicate what the red and blue arrows represent. Additionally, I did not find in the legend or methods what ratio of CHIP was added to the fractions in the right lanes, showing the disassembly into 70:CHIP, 90:CHIP, etc.

Reply: We referred to the red and blue arrow in the figure legend of Fig. 5b (page 34). Additionally we specified the applied molar ratios used for the control experiments also in the figure legend: *“Chaperone:CHIP(:Tau) molar ratios of 1:1(:5) were used for control experiments.”*

Reviewer #2:

This is an interesting manuscript, which explores the biophysical interactions of tau with the Hsp70-HOP-Hsp90 complex. This chaperone complex has been best associated with folding and stabilization of steroid hormone receptors and kinases. In addition, the individual factors Hsp70 and Hsp90 have been extensively studied for their effects on tau aggregation and microtubule binding. In brief, the authors have combined those two ideas here – using NMR and X-linking MS to study the binding of tau to the Hsp70-HOP-Hsp90 complex. Two of the most important findings are that p23 promotes tight binding, at least partially by binding directly to tau, and the ability of CHIP to displace the complex. All together, this is an important work, as it extends previous studies on the individual, binary interactions and begins to probe the impact of larger systems (likely to increasingly simulate the endogenous machines).

1. It is a little surprising that nucleotide is not required for these interactions, as previous studies using the GR system have shown that ATP hydrolysis is involved in assembly and progression of the Hsp70-HOP-Hsp90-p23 complex (Kirshke 2014 Cell; Noddings 2021 BioRxiv). The author's data is pretty clear, but this stark contrast with previous systems should be expanded upon in the discussion to make it clear that the tau system seems to have distinct features.

Reply: We thank the referee for raising this issue. The finding that nucleotides are not required for this interaction (see Supplementary Fig. 2e) indeed posed the question whether the ATP hydrolysis activity of Hsp70/Hsp90 plays any role for the chaperoning of Tau. We now reason this finding in a separate paragraph in the discussion section arguing that Tau binding serves for protein holding rather than protein folding (please see page 14 of the revised manuscript): *“So far, the Hsp70/Hsp90 multichaperone machinery is suggested as a promoter of protein folding and activation.³¹ For globular proteins, the chaperones are required to obtain distinct nucleotide states for client processing.^{36,38,39} Our finding that nucleotides are not required for Tau chaperoning is however in agreement with previous results that, for the substrate Tau, the affinity for Hsp90 remained unaffected with or without ATPyS.⁴⁰ Moreover, we found that the Tau interaction with the Hsp70/Hsp90 chaperone machinery was independent of the co-chaperone Hsp40 (Supplementary Fig. 1d), which is known to assist for the Hsp70:substrate interaction by stabilizing the chaperone's ADP-state.⁶ Hence, with regard to Tau as an intrinsically disordered protein, the interaction with the Hsp70/Hsp90 multichaperone machinery might be because of protein holding rather than protein folding, which in turn does not necessarily require energy.”*

2. The lack of nucleotide dependence also begs the question of how the complex is dis-assembled (and, therefore, regulated)? The effects of CHIP are interesting, but even this factor was poorly able to act on the p23-containing complex. One wonders if nucleotide exchange factors, such as BAG3 (Kirshke 2014), can reverse the complex assembly – as seen in the GR and kinase complexes?

Reply: We thank the referee for these considerations. Following our response to the referee's previous comment, we argue that the dissociation of the Hsp70/Hsp90 machinery: Tau interaction is likely to be driven by factors other than ATP hydrolysis. One of these factors could definitely be BAG3 but also other co-chaperones (e.g. Aha1 or PPlases, which modulate Hsp90 closing), could possibly contribute to the dissociation of the Tau: machinery complex. This has now been highlighted in the revised version of the manuscript (page 15): *“Instead of the event of ATP hydrolysis that is reported to complete the Hsp90 action for other substrates,⁸ additional co-chaperones such as Aha1 or PPlases could, similarly to what has been shown with CHIP (Fig. 5), regulate the dynamic assembly and disassembly of the Hsp70/Hsp90 chaperone machinery: Tau complex.”*

3. Similarly, another Hsp90 co-chaperone, Aha1, has been especially linked to tau proteostasis and the progression to aggregation (Shelton et al. 2017 PNAS; Criado-Marrero et al. 2021 Acta Neuropath. Commun; Singh et al. 2020 Cell Chem Biol.). If feasible, studying the effects of Aha1 on the dissolution of the Hsp70-HOP-Hsp90-tau complex would be interesting to add. Please note that this comment and the previous one are acknowledged to be a substantial amount of work.

Reply: This is indeed another genuinely excellent issue raised by the referee. However, we are convinced that this goes beyond the purposes of this manuscript, because it primarily focuses on the characterization of the interaction between Tau/PTau and the Hsp70/Hsp90 multichaperone machinery. The dissociation experiment with CHIP (Fig. 5) was conducted in order to show that the Tau:machinery interaction can be resolved at all, which we believe allows the interaction to be considered highly relevant also *in vivo*. Certainly, additional insights into the Tau:machinery interaction are highly worthwhile, and we believe that the current manuscript provides an excellent basis to address these questions in the future.

4. The co-factor free tau aggregation method is well-used here, but it is not clear if the polytetrafluoroethylene beads used to induce aggregation might similarly aggregate chaperones? This seems like a critical control.

Reply: We infer from this referee's comment that perhaps it might not have been clearly shown in the manuscript that in the aggregation experiment the (co-)chaperones alone do not aggregate into ThT-positive species. Indeed we show this control experiment in the supplementary data (please see Supplementary Fig. 2g, left graph). For clarity, we re-emphasized the respective part in the results section (page 8): "*In the conditions of the co-factor-free aggregation assay, only Tau but none of the chaperones or co-chaperones formed Thioflavin-T-positive amyloid fibrils (Supplementary Fig. 2g).*"

5. The ability of CHIP to displace the Hsp70-HOP-Hsp90-tau complex is interesting, but also somewhat unexpected. The affinity of CHIP and HOP for the EEVD motifs of Hsp70 and Hsp90 are (roughly) similar; so CHIP's ability to drive such complete dissolution of the complex at 1:1 stoichiometry is unexpected given the avidity expected within the pre-existing, large complex. What is the kinetics of this effect? One would expect it to be slow, as dissociation of the complex would seem to be required. An alternative is that CHIP binds directly to tau (Kim et al. 2021 Chem Sci 12:5599), so maybe this interaction also serves to assist in dissolving the complex?

Reply: We thank the referee for raising this issue. We demonstrated that CHIP is able to dissociate the Hsp70/Hsp90 chaperone machinery:Tau complex at an equimolar ratio (Fig. 5). We believe that this is not fully unexpected, because of the (roughly) similar affinities of Hop and CHIP for the chaperones. However, despite the roughly comparable affinities of Hop and CHIP for Hsp70/Hsp90 (Stankiewicz, M., et al. doi:10.1111/j.1742-4658.2010.07737.x (2010)), we note that CHIP has rather low intracellular concentrations (0.0094 μ M) when compared to Hop (1.2 μ M) (Kundrat, L. & Regan, L., doi:10.1021/bi100386w (2010)). It therefore remains to be elucidated to which extent a dynamic interplay between Tau retention *via* Hop and Tau degradation *via* CHIP is maintained by Hsp70 and Hsp90 *in vivo* or whether other co-chaperones are involved in the disassembly of the Tau:machinery interaction. We now also included these considerations into the results section (please see page 11).

Reviewer #3:

The details of interaction between Tau with Hsp90 have been investigated by combining NMR approach with small angle X-ray scattering (SAXS) and described in the author Zweckstetter's publication (Cell 2014, 27; 156(5): 963-74). In this reviewing paper, the authors observed the complex Hsp70:Hop:Hsp90:Tau and Hsp70:Hop:Hsp90:Tau:p23 formed on the native gels and interaction between proteins were detected by 900 MHz spectrometer NMR and crosslinking mass spectrometry. Overall, the results are compelling to support the authors' major conclusions. I have the following points about the manuscript.

1. The iBAQ method cannot provide a protein precise ratio of 1:1:1:1. The figure S1b shows the four proteins were identified in the complex of Hsp70:Hop:Hsp90:Tau. It is better to list the iBAQ intensity value obtained from the MaxQuant protein table than to show it on a figure.

Reply: We thank the referee for raising this issue. We agree that by means of the iBAQ method, it is not possible to make an absolutely precise statement about the molar ratio of the proteins of a complex. For this reason, we have omitted a conclusion about a molar ratio based on the iBAQ values. In addition, we have removed the graph previously shown in the Supplementary Fig. 1b and included a protein table listing the MaxQuant values as requested by the referee. These data can now be found in the Supplementary information (new Supplementary Table 1).

2. The stoichiometry of the Tau complex with the other three proteins in 2Hsp70:1Hop:2Hsp90 and (1Hsp70:1Hop:2Hsp90:1Tau)₂ complex has been defined by the density gradient centrifugation followed by size exclusion chromatography.

2Hsp70:1Hop:2Hsp90: 380kd (415kd by sequence calculation)

(1Hsp70:1Hop:2Hsp90:1Tau)₂: 853kd (739kd by sequence calculation)

There is a large mass difference between theoretical calculation and measurement for (Hsp70₁:Hop₁:Hsp90₂:Tau₁)₂ even considering Tau's apparent mass different with actual mass (Ref. Acta Neuropathol 2017, 133:665–704). May it be a mixture of various ratio species? Does the measurement method have limitation? Can the authors make an explanation?

Reply: We fully agree with the reviewer that size exclusion chromatography is only an approximate method to determine molecular weights and does not represent an absolute mass determination. We have therefore modified the text on page 10 describing the estimation (not absolute mass determination) of the sizes of the complexes. Indeed, we observe a mixture of complexes with various stoichiometries, as shown in Supplementary Figure 3b. However, there appears to be an enrichment of a "main" high-molecular weight complex with fixed stoichiometry (appearing as the darkest band above 200 kDa in the SDS page gel, both in the 70:Hop:90:Tau system and the 70:Hop:90:Tau:p23). Unfortunately, because the "main" complex co-elutes with other complexes and non-crosslinked proteins in SEC, the mass determination by elution volume is not accurate. We therefore revised the text as follows (page 10): *"For the Hsp70:Hop:Hsp90 complex in the absence of Tau, we obtained an elution volume corresponding to a molecular mass of 411 kDa (Fig. 4e, f). In the presence of Tau, however, the estimated mass increased to 739 kDa (Fig. 4e, f), i.e. much larger than the molecular weights expected for a Hsp70₂:Hop₁:Hsp90₂:Tau₁ complex (425 kDa) and a Hsp70₁:Hop₁:Hsp90₂:Tau₁ complex (354 kDa). However, given the broad nature of the complex peak in SEC it cannot be excluded that multiple stoichiometries ranging from ~400 kDa to ~700 kDa may be formed (corresponding to elution volumes ~11 mL to ~9 mL). This is consistent with SDS-PAGE analysis of the cross-linked Hsp70:Hop:Hsp90:Tau complexes that show band smearing corresponding to molecular weights above 200 kDa (Supplementary Fig. 3b). The wide range of stoichiometries may correspond to 1:1 to 2:2 complexes of Tau:machinery, with possibly one or two Hsp70 units within the Hsp70:Hop:Hsp90 machinery. Notably, in the presence of p23 the main cross-linked complex shows a quite defined band in SDS-PAGE (Supplementary Fig. 3b) as well as a narrower peak in the SEC chromatogram (~700 kDa molecular weight corresponds to the elution volume at the peak maximum, Fig. 4e). This suggests that the addition of p23, which directly binds to Tau (Fig. 2d), further stabilized*

the high-molecular weight complex. The combined data point to the formation of high molecular complexes of Hsp70:Hsp90:Tau and Hsp70:Hsp90:Tau:p23 with variable stoichiometry.

3. The complex assembly in Fig 4h are derived from the EM results of references [1-3] without any other modelling experiment results support e. g. crosslinking data with homology model from EM. Crosslinking mass spectrometry data have no contribution to solving this complex assemblies.

Reply: We thank the referee for raising this issue. We agree that no final structure can be deduced on the sole basis of the data from cross-linking followed by LC-MS/MS and have therefore removed the complex assembly in Fig. 4h, also acknowledging the fact that the stoichiometry of the assembly is variable.

4. Questions following the Crosslinking experimental data

a. This is a vitro crosslinking experiment in an aqua-buffer. The authors use a non-water soluble crosslinker DSS instead of the water soluble partner BS3. Why?

Reply: Prior to the cross-linking analysis in large scale we performed a titration experiment in order to define the appropriate cross-linker concentration. In the case of DSS, we obtained distinct bands on the top of the SDS gel indicating the formation of distinct high molecular weight complexes. In the case of BS3, no distinct bands but rather a smear all over the gel was obtained. We therefore decided to further work with DSS as cross-linker.

DSS was initially solubilized in DMSO and then added to the mixture. We did not observe any precipitation upon the addition of DSS. This is now mentioned in the Methods section (page 20).

b. Either DSS/BS3 or EDC may not be a good choice for unstructured Tau. NHS ester crosslinkers capture a lot of flexibility interaction (protein kinetics trapping). Photo-crosslinker e. g SDA is a better choice for detecting core interactions (J Proteome Res, 18 (2019), pp. 934-946).

Reply: We chose DSS or EDC in order to capture the interactions of Lys-Lys and Lys-Glu/Asp amino acid residues within the proteins. These residues are distributed along the entire sequence of Tau. While DSS has a certain spacer length and might capture interactions in more flexible regions, EDC is a “zero length” crosslinker and – similar to SDA – may not capture all the interactions of the more flexible protein regions; rather, it should preferentially capture those that are in very close spatial proximity. Thus, we are very confident that the combination of these two crosslinkers has allowed us to capture a variety of crosslinks that reveal flexible and core interactions. Since the pattern of crosslinked residues obtained by DSS or EDC is similar, we are convinced that the use of a further crosslinker is unlikely to provide essential additional information for the unstructured Tau.

c. Peptide size exclusion (pSEC, Superdex Peptide 3.2/300 column on an ÄKTAmicro system, GE Healthcare) was performed to enrich for crosslinked peptides. The raw data showed crosslinking peptides distributed cross at least 10 fractions and have not been enriched properly (Normally crosslinking peptides are enriched at 3-4 fractions on this column)

Reply: Thanks for raising this issue. Indeed, for the analysis of crosslinked peptides in the original manuscript a relatively large number of pSEC fractions were used. However, we do not think that there was a severe issue with the enrichment of cross-linked peptides as supported by the plots shown below. The plots display the number of identified crosslinked (CSMS) and non-crosslinked (PSMS) peptides in the individual fractions of the pSEC in a database search with pLink and MaxQuant. The elution profile shows that the crosslinks peaked at fractions ~22–26 for both crosslinkers. In addition, there is a clear separation of crosslinked (CSMS replicates R1 and R2) and linear peptides (PSMS replicates R1 and R2) in pSEC. On the basis of the referee's concerns we further re-analyzed our data with 1% FDR, 10 ppm MS1 and 20 ppm for MS2 from fractions #21–28 (DSS) and #20–28 (EDC). See also below, points d and e.

d. The crosslinked samples for both DSS and EDC were overloaded and separated at a very short LC gradient (46.5min gradient in a 58 minute run) on LC-MS. There are a lot of peptides co-elution without proper separation and that has caused the crosslinks identified all over the sequences to contain a high percentage of false positives. The crosslinking data shown in Figure S4c are a part of crosslinks by 5% FDR plus high score cut off (which have not shown all the crosslinks that pass 5% FDR threshold). I identified around 2500 links with 1% FDR for duplications of EDC data sets with a setting of MS1 error 3 ppm and a MS2 error of 10 ppm in combined fraction 25-30).

Reply: Thanks for raising this issue. We have re-analyzed the crosslinking data and adapted the manuscript accordingly wherever necessary. For crosslinking analysis with pLink we have now used fractions #21–28 (DSS) and #20–28 (EDC) with 1% FDR exclusively with 10 ppm for MS1 and 20 ppm for MS2. Moreover, we only considered crosslinked peptides that meet these criteria and were identified with ≥ 3 CSMs. We did not apply any score cut-off in pLink. Such criteria lead to a confident identification of crosslinks. We further note that the Urlaub laboratory in general does not use top fractions of crosslinks only, but also uses some of the preceding fractions as these still contain crosslinks (see above, point c). We have modified all relevant Figures and Supplementary Tables accordingly (please see below, point e).

Furthermore, we have compared the crosslinks at 1% FDR with 10 ppm MS1 and 20 ppm MS2 in a search with 1% FDR and 3 ppm MS1 and 10 ppm MS2 also with 1 % FDR (as suggested by the reviewer). The figure below shows a comparison between the numbers of crosslinks obtained for DSS and EDC with these settings. The views at the top show the xiView analysis of DSS and EDC inter-crosslinks. The number of DSS inter-crosslinks is reduced with 3 ppm at MS1, but such reduction in CSMSs numbers does not lead to any significant change in the overall pattern of inter-protein crosslinks, i.e. in the regions of the different proteins that crosslink to each other. In the case of EDC almost no change in terms of numbers or pattern is detectable. The different values are summarized in the table at the bottom of the figure.

Top: xiView of inter crosslinks for DSS and EDC at 10 ppm and 3 ppm precursor mass tolerance in MS1. The color code corresponds to the number of CSMs.

Bottom: Table summarizing the pLink analysis. all: all CSMs identified with DSS and EDC with 10 or 3 ppm precursor mass tolerance in MS1; unambiguous: CSM without ambiguous CSMs, i.e. those that match not only to one protein; unique CSMs: Unique crosslinking sites; selected: unique CSMs with ≥ 3 CSMs of intra- and inter-crosslinks.

We would also like to note that we observed a certain mass deviation in our mass spectrometer that made it necessary to choose wider settings in our searches. Below we show plots of the mass error of linear and DSS or EDC crosslinked peptides. This mass deviation prompted us to use less stringent search settings, i.e. 10 ppm in MS1. Nevertheless, our search settings used for the analysis in our manuscript (10 ppm MS1 and 20 ppm MS2) are still more stringent than the original proposed standard search settings from pLINK (20 ppm MS1 and 20 ppm MS2; Chen et al., A high-speed search engine pLink 2 with systematic evaluation for proteome-scale identification of cross-linked peptides).

e. The crosslinking data (selected by 5% FDR then extra score cut off with at least three times of identification) were validated by five PDBs of Hsp90, Hsp70-NBD, Hsp70-SBD, Hop-TPR1, Hop-TPR2A, and p23. Only 64 % of the selected EDC links are within the distance constraints. Could the authors discuss why the percentage is so low?

Reply: We have re-analyzed the crosslinking data derived from the pSEC fractions #21–28 (DSS) and #20–28 (EDC) with an FDR of 1%, 10 ppm MS1, 20 ppm MS2 (see above, point d). 10 ppm on MS1 and 20 ppm are the standard settings used in the Urlaub laboratory. As shown above (point d) the search with a narrower mass deviation window did not significantly change the overall pattern of detected crosslinks but mainly the numbers of CSMs detected. Reducing the FDR from 5% to 1% does change the data quality. Therefore, we changed all Figures (Fig. 4c, Supplementary Fig. 3 and the Supplementary Table 2 and 3) according to the new search parameters. Importantly, we only considered crosslinks that were identified under these particular settings with ≥ 3 CSMs. Plots for all the proteins were modified accordingly in Fig. 4c and Supplementary Fig. 3. Validation of crosslinks – e.g., on the available Hsp90 structures – showed that the numbers of crosslinks of Hsp90 that violate the constraint rule, i.e. those that exceeded 30 Å, were diminished from 40% to 28% (Supplementary Fig. 3). We suggest that the reason why there is still a number of crosslinks that exceed the distance constraints might be due to larger scale structural changes in the protein complex.

f. The crosslinked products were separated by sucrose gradient purification. Do the authors have any SDS gel image of crosslinked product bands? Is it a mixture of different complexes and sub-complexes?

Reply: We ran the respective SDS gel after the sucrose density gradient centrifugation, and this gel is now shown in the updated version of Supplementary Figure 3b:

Top: SDS-page analysis (4–15 % gradient gels; BioRad) of the cross-linked Hsp70:Hsp90:Tau:p23 complex after sucrose density gradient purification (10–25 % sucrose). The fraction of interest is highlighted with a grey square and used for the subsequent SEC run.

Bottom: SEC analysis of the fraction highlighted on top (SD200 10/30).

Purification by density gradient centrifugation appeared of high efficiency, as most of the unbound proteins were retained in the upper fractions (left side of the gel). Thus, the fraction of interest predominantly contained the cross-linked complex referring to the top band in fraction 12. In the updated manuscript, we also refer to Supplementary Figure 3b when discussing molecular weight determination by SEC (page 10) and explaining the stabilization of the complex by p23 (page 11).

5. The data deposit reference PXD024457 is the wrong identifier, the correct one is PXD027287

Reply: Thanks for spotting. Given the reanalysis of the MS data (see points 3 and 4), we have updated the identifier in the manuscript (page 26).

Reviewers' Comments:

Reviewer #1:

Remarks to the Author:

The revised manuscript contains several additional experiments that have substantially strengthened the conclusions, addressing a number of my concerns.

My major unresolved issue is regarding the published affinities for the tau-chaperone complex formation. As a major part of the paper discusses that the Hsp70/Hsp90 machinery binds tau with high affinity, experimental evidence supporting this claim must be provided.

While I completely understand the necessity of using native gels, as it was the only method that worked for the authors, this unfortunately still does not make it a quantitative approach. This is even more problematic, due to the fact that the results from the NMR experiments do not support these measurements, reporting a much weaker interaction.

The author's response to my initial comments regarding this issue does not address the issue raised. The authors reply that "the NMR signal intensities of P do not accurately represent the population of unbound P since lines are broadened by the exchange between free and bound states of P." This statement is of course absolutely correct, however this would mean that upon binding of the chaperone machinery to tau, one should observe a reduction in the free Tau peaks (P) intensity even greater than the population of the bound state. The reduction in intensities for Tau upon addition of chaperones is, however, much smaller than the expected % of complex formation, based on the reported k_{ds} . For example, in figure 1f and 2c the NMR signal of tau decreases by ~40%, which indicates that less than 40% of tau is bound to the chaperone complex. This is, however, entirely inconsistent with the reported k_d of 1.3 μ M, for which (based on the reported protein concentration) 96% of tau should be in complex with the chaperones. Furthermore, the updated figure 2c (bottom panel) shows that less than 20% of Hsp70 or Hsp90 are found in complex with tau, which is again, inconsistent with the reported k_{ds} of 4.1 μ M and 2.9 μ M, respectively.

The authors also refer to the manuscript by Borgia et al., and state that "A particular striking example of the difficulty to directly associate ligand induced changes in the NMR spectra of an intrinsically disordered protein with the affinity of the interaction was provided in Borgia, A. et al., demonstrating affinity values in the pmol range and only small perturbations in NMR spectra." I would just like to point out that this refers to the general small changes in the chemical shifts, the size of which indeed does not report on the strength of an interaction. On the other hand, the intensities of the free protein (P) in Borgia et al. are zero, as all of the free protein was converted to the bound state, as should be expected for such a tight interaction.

Therefore, in my opinion, the authors must address this issue prior to publication. This can be done by either measuring the k_{ds} by an additional, more quantifiable method, or alternatively not reporting the k_{ds} at all in the manuscript. If the latter option is chosen, all references to tight affinity complexes should be removed.

In addition, there are also some minor comments -

- 1) The legend for figure 1e is missing
- 2) Line 213-214: The authors state that the P2 region was shown by NMR to be the additional binding site for Hsp90. This is not evident from the results presented in figure 2c, which show binding of p23 to the P2 region, but no such interaction with Hsp90 (the black plot on the bottom).
- 3) In figure 3, the authors show a structural model for Hsp70-Hop-Hsp90 complex. Could they

speculate how this structure would change upon the binding of tau?

4) Please provide the error bars for the molecular weight calculations in figure 4.

5) The aggregation profile of the hyper-phosphorylated tau is quite unusual. Could the authors elaborate on that in the text?

6) The conclusion that the chaperone complex interacts with tau in a nucleotide- and Hsp40-independent manner should be revised. While this might be true in the test tube, where ADP can be added as needed, in the cell Hsp70 is found in an ATP-bound state, and Hsp40s are needed to convert the chaperone to the ADP state, in which it can bind substrates (tau).

7)Supplementary figure 2 is listed as figure 1 in the legends

Reviewer #2:

Remarks to the Author:

The authors have made substantial clarifications, especially to the Discussion and Methods, which further improve the work.

Reviewer #3:

Remarks to the Author:

The authors have satisfactorily addressed my previous concerns. A revised version of this manuscript has been improved, so I support its publication in Nature Communications.

Response to Reviewer#1

The text highlighted in green correspond to the edited sentences in the main manuscript.

Reviewer #1 (Remarks to the Author):

The revised manuscript contains several additional experiments that have substantially strengthened the conclusions, addressing a number of my concerns. My major unresolved issue is regarding the published affinities for the tau-chaperone complex formation. As a major part of the paper discusses that the Hsp70/Hsp90 machinery binds tau with high affinity, experimental evidence supporting this claim must be provided. While I completely understand the necessity of using native gels, as it was the only method that worked for the authors, this unfortunately still does not make it a quantitative approach. This is even more problematic, due to the fact that the results from the NMR experiments do not support these measurements, reporting a much weaker interaction. The author's response to my initial comments regarding this issue does not address the issue raised. The authors reply that "the NMR signal intensities of P do not accurately represent the population of unbound P since lines are broadened by the exchange between free and bound states of P." This statement is of course absolutely correct, however this would mean that upon binding of the chaperone machinery to tau, one should observe a reduction in the free Tau peaks (P) intensity even greater than the population of the bound state. The reduction in intensities for Tau upon addition of chaperones is, however, much smaller than the expected % of complex formation, based on the reported k_{ds} . For example, in figure 1f and 2c the NMR signal of tau decreases by ~40%, which indicates that less than 40% of tau is bound to the chaperone complex. This is, however, entirely inconsistent with the reported k_d of 1.3 μ M, for which (based on the reported protein concentration) 96% of tau should be in complex with the chaperones. Furthermore, the updated figure 2c (bottom panel) shows that less than 20% of Hsp70 or Hsp90 are found in complex with tau, which is again, inconsistent with the reported k_{ds} of 4.1 μ M and 2.9 μ M, respectively.

The authors also refer to the manuscript by Borgia et al., and state that "A particular striking example of the difficulty to directly associate ligand induced changes in the NMR spectra of an intrinsically disordered protein with the affinity of the interaction was provided in Borgia, A. et al., demonstrating affinity values in the pmol range and only small perturbations in NMR spectra." I would just like to point out that this refers to the general small changes in the chemical shifts, the size of which indeed does not report on the strength of an interaction. On the other hand, the intensities of the free protein (P) in Borgia et al. are zero, as all of the free protein was converted to the bound state, as should be expected for such a tight interaction.

Reply: We are grateful to the reviewer for the careful examination of the affinity data presented in this manuscript. However, we still disagree that the %intensity reduction of NMR signal in Figure 1f and 2c is directly related to the fraction of Tau bound to the chaperone complex. We provide a more detailed explanation here: In the intermediate chemical shift time scale of NMR, the population of unbound Tau cannot be computed from the NMR signal intensities because the interconversion or "exchange" between bound Tau and unbound Tau occurs at a frequency comparable to the chemical shift difference between these two states. A consequence of this is the appearance of a "broader" NMR signal, which corresponds to a lower signal intensity (i.e. I/I_0 lower than 1.0). This is what is meant by our previous reply "the NMR signal intensities of P do

not accurately represent the population of unbound P since lines are broadened by the exchange between free and bound states of P.”

Please also note that in the article of Borgia et al. the interaction between the two proteins (ProTα and H1) also falls into the intermediate exchange time scale (Borgia et al. Figure 3g and Extended Data Figure 2). The authors show NMR titrations monitoring ¹⁵N-HSQC spectra of ¹⁵N-ProTα while incrementing H1, akin to what we present in this manuscript for ¹⁵N-HSQC spectra of ¹⁵N-Tau titrated with the chaperone machinery (specifically Figure 1e-f Tau with Hsp70:Hop:Hsp90, and Figure 2c-d Tau with Hsp70:Hop:Hsp90:p23). We disagree with the reviewer’s assessment that “the intensities of the free protein (P) in Borgia et al. are zero.” We believe that perhaps this assessment resulted from a slight confusion about how the NMR signal intensity (I/I₀) plots in Borgia et al. were presented. Upon closer examination of the NMR signal intensity plot in Figure 1 and Extended Data Figure 2 of Borgia et al., it is evident that the gaps in the plot (residues ~60-68) do not correspond to zero intensities of free protein. Rather they correspond to proline residues (indicated by light grey stars) which do not show H-N peaks in H-N correlation spectra and therefore do not have peak intensity values, residues omitted because they were unassigned (light grey stars), and residues omitted due to “insufficient data quality” (dark grey stars) as indicated in the caption of Extended Data Figure 2. For residues that do not have these issues, the signals of the free protein were not zero.

Therefore, in my opinion, the authors must address this issue prior to publication. This can be done by either measuring the kds by an additional, more quantifiable method, or alternatively not reporting the kds at all in the manuscript. If the latter option is chosen, all references to tight affinity complexes should be removed.

Reply: While we maintain our interpretation of the affinity regimes measured by native gels and complemented by NMR experiments shown in this manuscript, we acknowledge that the K_D values presented in our manuscript were not acquired by conventional quantitative experiments. Therefore, we have revised parts of the manuscript to clarify these points (page 4-5): “From the concentration-dependent intensity increase of the Hsp70:Hop:Hsp90:Tau complex band, we determined the apparent K_D = 1.3 ± 0.1 μM (Fig. 1d). Given that native gel-based affinity determination is semi-quantitative, the K_D values reported here should be regarded as apparent dissociation constants.”

In addition, we removed the statement “The affinity of Tau for the Hsp70:Hop:Hsp90 complex thus exceeds that for binding to the individual chaperones (Fig. 1d)” since this comparison could require more accurate K_D determination.

We now also point out that the equation used for K_D determination (page 18)

$$I = I_{max} \cdot \frac{\text{Tau}_{total}}{\text{Tau}_{total} + K_D}$$

is valid only when it is assumed that the binding stoichiometry of Tau:machinery is 1:1. As discussed in the manuscript (page 10 and Figure 4) our findings from SEC and SDS-PAGE analyses of the cross-linked Tau:machinery complexes show a wide range of stoichiometries. Accordingly, we state on page 18 under “Affinity determination”: “In either case, the dissociation equilibrium constant K_D was determined with the equation

$$I = I_{max} \cdot \frac{\tau_{total}}{\tau_{total} + K_D}$$

using the solver function in Excel v16.43. This equation is based on the simplified assumption that the binding stoichiometry of Tau to the chaperone machinery is 1:1.

In addition, there are also some minor comments –

1) The legend for figure 1e is missing

Reply: Thanks for spotting. The figure legend has been included into the upper right spectrum (1:0.2, 1:1, 1:2) and is now described in the figure caption: “E) 2D 15N-1H HSQC spectra of Tau alone (grey) and with increasing concentrations of the Hsp70:Hop:Hsp90 complex (1:1:1 molar ratio) using Tau:machinery molar ratios of 1:0.2 (purple), 1:1 (pink) and 1:2 (red).”

2) Line 213-214: The authors state that the P2 region was shown by NMR to be the additional binding site for Hsp90. This is not evident from the results presented in figure 2c, which show binding of p23 to the P2 region, but no such interaction with Hsp90 (the black plot on the bottom).

Reply: The broadening (meaning the observation of I/I_0 values less than 1.0) involving part of the P2 region in Figure 2c indicates that this is a possible interaction site for Tau-Hsp90 binding. To clarify this we show below the revised version of Figure 2c in which the highlighted part in grey corresponds to residues in P2 which show I/I_0 values <1.0 in the red, blue, and black bar plots but $I/I_0 \sim 1.0$ in the grey (Tau:70) plot:

In the latter half of the P2 region (residues ~221-240), I/I_0 values are <1.0 for the black plot (Tau:90) but $I/I_0 \sim 1.0$ for the grey plot (Tau:70) indicating that this region could be a binding site for Hsp90 and not for Hsp70.

To emphasize this distinction we have revised the statements on pages 9-10: “In agreement with a direct Tau:p23 interaction demonstrated by NMR (Fig. 2d), the majority of p23 cross-links were found with the P2 region of Tau (Fig. 4c, d and Supplementary Fig. 3e). Notably the second half of the P2 region (residues ~220-240, highlighted in grey) was suggested by NMR to be an additional binding site for Hsp90 that is not involved in Hsp70 binding (Fig. 2c). This is suggested

by the peak broadening of these Tau residues upon titration with Hsp90 ($I/I_0 < 1.0$), which is largely absent in the titrations with Hsp70 (most I/I_0 values in this region are close to 1.0).”

3) In figure 3, the authors show a structural model for Hsp70-Hop-Hsp90 complex. Could they speculate how this structure would change upon the binding of tau?

Reply: Please note that Figure 3c is a cartoon representation for the Hsp70-Hop-Hsp90 complex. Owing to the current lack of structural evidence for the Hsp70-Hop-Hsp90-Tau complex as well as the relative orientations of the machinery components (this point was also raised by referee #3), we believe that it would not be prudent to speculate on how the Hsp70-Hop-Hsp90 could change upon the binding of Tau. We however fully agree that providing structural insights for this complex assembly is of high interest and could be addressed in future investigations.

4) Please provide the error bars for the molecular weight calculations in figure 4.

Reply: We have revised Figure 4 to include the molecular weight reported as average +/- standard deviation.

5) The aggregation profile of the hyper-phosphorylated tau is quite unusual. Could the authors elaborate on that in the text?

Reply: We have inserted the following explanation on page 13 of the main text:

“Under our heparin-free aggregation conditions, both Tau and pTau aggregate in a sigmoidal fashion in the absence of chaperone machinery components (Fig. 2e and Fig. 6f). Noticeably, for pTau the aggregation profile appears more complex in the presence of chaperones.

Bell-shaped ThT curves were observed for pTau-70Hop90 and pTau-70Hop90p23, and a wave-like pattern was shown for pTau-90 (Fig. 6f). These unusual ThT curves were not observed for Tau with chaperone machinery components, which suggests that alternate aggregation pathways that resulted in amyloid-deficient pTau assemblies were promoted by the interaction between pTau and the chaperone machinery components. Under these conditions it is possible that pTau formed ThT-negative oligomeric aggregates or fibril polymorphs that led to the fluctuations in ThT intensity.

6) The conclusion that the chaperone complex interacts with tau in a nucleotide- and Hsp40-independent manner should be revised. While this might be true in the test tube, where ADP can be added as needed, in the cell Hsp70 is found in an ATP-bound state, and Hsp40s are needed to convert the chaperone to the ADP state, in which it can bind substrates (tau).

Reply: Thanks for this suggestion. To avoid any misinterpretation, we now state on page 15: “Our finding that nucleotides are not required for Tau chaperoning is however in agreement with previous results that, for the substrate Tau, the affinity for Hsp90 remained unaffected with or without ATPyS.⁴⁰ It remains to be seen how the Tau-machinery interaction in the cell (with a pool of nucleotides in the cytosol) differs from what we observed in the absence of nucleotides in vitro.”

7) Supplementary figure 2 is listed as figure 1 in the legends

Reply: Thanks for spotting. We have submitted a revised version of the supplementary information that incorporates this correction and includes source data pertaining to the supplementary figures.